# Measurement report: Number size distribution of sub-40 nm particles in the Amazon rainforest

Jianqiang Zhu[1, a, ★], Guo Li[2, 1, ★], Uwe Kuhn[1, a], Bruno Backes Meller[3], Christopher Pöhlker[1], Paulo Artaxo[3], Ulrich Pöschl[1], Yafang Cheng[4], Hang Su[2, 1]

1Multiphase Chemistry Department, Max Planck Institute for Chemistry, 55128 Mainz, Germany
2State Key Laboratory of Atmospheric Environment and Extreme Meteorology, Institute of Atmospheric Physics, Chinese Academy of Sciences, Beijing 100029, China
3Institute of Physics, University of São Paulo, São Paulo 05508-900, Brazil
4Aerosol Chemistry Department, Max Planck Institute for Chemistry, Mainz 55128, Germany
[a] now at: Aerosol Chemistry Department, Max Planck Institute for Chemistry, Mainz 55128, Germany
★ These authors contributed equally to this work.

*Correspondence to*: Hang Su (suhang@mail.iap.ac.cn)

**Abstract:**

The Amazon rainforest is a unique environment to investigate aerosol properties with limited impact from human
activities, further providing a new perspective to look at the aerosol characteristics in regions heavily affected by anthropogenic emissions. Obtaining the size distributions of nucleation mode particles in the atmosphere is key to understanding aerosol formation, evolution and their impacts. Although routine and long-term aerosol measurements have been conducted in the Amazon region, information regarding sub-10 nm particles is still limited. In this study, we performed aerosol measurements from December 2022 to January 2023 on a 54-meter-
high platform of the Amazon Tall Tower Observatory (ATTO). Three advanced instruments namely the Nano Condensation Nuclei Counter (nCNC), the Neutral Cluster and Air Ion Spectrometer (NAIS), and the Nano-Scanning Mobility Particle Sizer (SMPS), were employed to measure the number size distributions of aerosol particles and naturally charged ions smaller than 40 nm. The results reveal that the median total number concentration of the measured particles with diameters ranging from 1.5 nm to 1000 nm was 969 $cm^{-3}$. We found
a large number of particles smaller than 3.5 nm, which accounted for up to 59 % of the measured total number concentration. There was a significant increase in the number concentration of sub-3 nm particles in January 2023 (median, 573 $cm^{-3}$) compared to December 2022 (371 $cm^{-3}$). The median number concentration of particles above 3.5 nm in December and January were 481 and 335 $cm^{-3}$, respectively. No typical regional new particle formation events were observed throughout the measurement period. However, clear diurnal variations were observed for
the sub-3nm particles under pristine conditions, with the maximum concentration around noontime. Similar diurnal patterns were also observed for natural cluster ions (0.8-2 nm), with their concentration in January slightly higher than in December. Quantifying the properties of the aerosol particles in the Amazon rainforest helps to understand the processes governing the aerosol budget in the pristine atmosphere, and is essential for determining the impact of anthropogenic aerosols on climate.

## 1 Introduction

Atmospheric aerosol particles significantly impact atmospheric chemistry and the Earth's climate directly and indirectly (Seinfeld and Pandis, 2016; Ipcc, 2022). Aerosols affect the atmospheric chemistry by directly serving as reactive surfaces/matrixes to facilitate and alter chemical reactions in the atmosphere (Pöschl and Shiraiwa, 2015; Farmer et al., 2015). Heterogenous and multiphase reactions occurring on aerosols can be an important source/sink for many critical gas species (Su et al., 2020; Ravishankara, 1997). Meanwhile, aerosols can influence the Earth's radiation balance by absorbing and scattering sunlight, which modulates the energy reaching the ground (Charlson et al., 1992; Myhre, 2009). In addition, by acting as cloud condensation nuclei (CCN), aerosols impact cloud formation and precipitation patterns, affecting the Earth's climate indirectly (Williamson et al., 2019; Merikanto et al., 2009; Rosenfeld et al., 2008).

Probing the basic properties of aerosols, such as their size distribution, is key to understanding their impacts. The size distribution of atmospheric aerosol particles is typically characterized by several modes: nucleation mode (particles with diameter smaller than ~25 nm), Aitken mode (from ~25 to ~100 nm), the accumulation mode (from ~100 to ~1000 nm), and coarse mode (particle with diameter larger than 1000 nm) (Whitby, 1978; Seinfeld and Pandis, 2016). With the development of instrumentation and the growing interest in the formation of new particles, more and more measurements were used to investigate the properties of nucleation mode particles in the last decade (Kulmala et al., 2013). However, global observations of particles below 10 nm and particularly sub-3 nm aerosols are still challenging, because these small particles are bearing high diffusive loss, low charging efficiency, and unknown chemical compositions (Stolzenburg et al., 2023; Kangasluoma et al., 2020; Kangasluoma and Attoui, 2019). With the rapid development of aerosol measurement techniques, some advanced instrumentation such as the diethylene glycol-based electrical mobility spectrometer (DEG-SMPS), the particle size magnifier (PSM), and the neutral cluster and air ion spectrometer (NAIS) have made the characterization for the sub-3 nm aerosols possible (Jiang et al., 2011; Vanhanen et al., 2011; Mirme and Mirme, 2013). These instruments have been widely used in several field measurements in recent years (Huang et al., 2024; Hong et al., 2023; Gonzalez Carracedo et al., 2022; Olin et al., 2022; He et al., 2021; Kangasluoma et al., 2020; Baccarini et al., 2020).

The Amazon rainforest is a globally important ecosystem and plays a major role in the climate system (Andreae et al., 2015). Measurements of particle number size distribution (PNSD) in the Amazon have been conducted during several field campaigns over the past several decades (Artaxo et al., 2022; Martin et al., 2016; Andreae et al., 2015; Martin et al., 2010; Rissler et al., 2006; Guyon et al., 2005; Rissler et al., 2004; Krejci et al., 2003; Zhou et al., 2002; Andreae et al., 2002). Seasonal variations of emissions and meteorology significantly alter the size distribution of submicron particles in the Amazon (Artaxo et al., 2022). During the wet season, Aitken and accumulation modes contribute almost equally to the particle size spectrum, with particle number concentrations ranging from 300-400 cm$^{-3}$ in the Central Amazon (Franco et al., 2022; Rizzo et al., 2018; Martin et al., 2010; Zhou et al., 2002). Biogenic sources and processes are dominant in the wet season, and the composition of the Amazon atmosphere approximates pre-industrial conditions during certain periods (Pöhlker et al., 2018). During the dry season, the particle number concentration is around 1000-1400 cm$^{-3}$, and their size distribution is mainly dominated by the accumulation mode (Artaxo et al., 2022; Rizzo et al., 2018). This is mainly influenced by local biomass burning emissions and long-term transport (Artaxo et al., 2022; Holanda et al., 2023; Andreae et al., 2018).

Nucleation mode particles were infrequently observed in the planetary boundary layer (PBL) of the Amazon, suggesting that new particle formation (NPF) events rarely occur (Artaxo et al., 2022; Rizzo et al., 2018; Andreae et al., 2015). This is different from other forest regions, such as the boreal forest of Hyytiälä (Finland), where typical NPF days account for 10 to 40 % of all measurement days (Kulmala et al., 2022; Kerminen et al., 2018). Based on multi-year measurements on a site (2.594° S, 60.209° W) located 60 km northwest of Manaus, NPF and apparent growth event days accounted for only 3% of the total measurement periods (Rizzo et al., 2018). However, no NPF events were observed at ground level inside the rainforest site (2.609° S,60.209° W) (Wimmer et al., 2018). The same study conducted on an open pasture site in the Amazon region observed the NPF events, and authors linked the NPF event to the pollution plume from Manaus (Wimmer et al., 2018). Previous observations in the Amazon were technically restricted to measure particles larger than 10 nm only, and occasionally observed particle growth starting from about 20 to 40 nm commonly related to downdrafts (Franco et al., 2022; Rizzo et al., 2018). The source of these particles, however, remains unclear. One of the possible sources is vertical transport during precipitation events, where ultrafine particles formed in the upper troposphere and were subsequently injected into the PBL by convective downdrafts (Wang et al., 2016; Andreae et al., 2018; Machado et al., 2021).

Despite that, Machado et al. (2024) recently analyzed aerosol and trace gas changes during downdrafts and attributed the characteristic particle bursts to boundary layer NPF, driven by a sudden increase in ozone and decrease in the condensation sink.

In order to fill the gap in measurements of particles and ions smaller than 10 nm in the Amazon region, we performed comprehensive measurements on the Amazon Tall Tower Observatory (ATTO) tower from 29 November 2022 to 26 January 2023. This measurement has been part of the Chemistry of the Atmosphere: Field Experiment in Brazil (CAFE-Brazil) campaign, designed to study tropospheric oxidant photochemistry in combination with particle formation and growth mechanisms under clean conditions over the Amazon rainforest (Halo-Research, 2022). This study presents the results of the fundamental properties of particles and ions obtained from a suite of advanced aerosol instruments. The main objectives of the present study are: (1) to characterize the number size distribution of particles, especially in the sub-10 nm size range; (2) to elucidate the life cycle dynamics of aerosol particles by investigating their diurnal variability in different size ranges; (3) to assess the potential occurrence of NPF events in the Amazon rainforest by examining the formation and growth properties of nucleation mode particles.

## 2 Methods

### 2.1 Field Measurement

#### 2.1.1 ATTO site

The measurements were conducted on a 54-meter-high platform of the Amazon Tall Tower Observatory (ATTO), situated at the center of the Amazon Basin (2.1459° S, 59.0056° W, 130 meters above sea level). The platform is about 20 m higher than the forest canopy. The Amazon Basin features minimal large-scale relief, but a dense drainage network has resulted in the formation of a landscape comprising plateaus and valleys on a smaller scale (Andreae et al., 2015). The region comprises a variety of forested ecosystems, with non-flooded upland forests predominantly found on the plateaus (Artaxo et al., 2022). As a key research site, ATTO is of great significance for comprehensive studies of atmospheric processes and their complex interactions with climate and ecosystems (Andreae et al., 2015; Pöhlker et al., 2019). The height of the platform enables the measurement of aerosols above the canopy level. The nearest city, Manaus, is located approximately 150 km southwest and downwind of the tower. For parts of the year with minimal human influence, the atmosphere provides clean continental background conditions. Further details regarding the ATTO site can be found in Andreae et al. (2015).

#### 2.1.2 Measurement periods

The field campaign took place from 29 November to 18 December 2022 and from 2 January to 26 January 2023, with a total of 20 measurement days in November and December, and 25 days in January. According to the definition of Amazonian seasons, the dry season lasts from August to November and the wet season from February to May (Pöhlker et al., 2016; Andreae et al., 2015). Therefore, the measurements carried out in December and January represent the end of the dry season and the beginning of the wet season, respectively.

### 2.2 Instrumentation overview

The measurement setup consisted of an ensemble of complementary instruments designed to quantify ions and neutral aerosol particles with diameters ranging from 1 nm to 40 nm. This ensemble included a Nano Condensation Nucleus Counter (nCNC, model A11, Airmodus, Finland), a Neutral cluster and Air Ion Spectrometer (NAIS, Airel, Estonia), and a Nano-Scanning Mobility Particle Sizer Spectrometer (SMPS, model 3938, TSI, USA). The status of each instrument was regularly monitored on a daily basis throughout the entire campaign. All the inlet lines, except for NAIS, were positioned horizontally without bends. In addition, a rain cover was installed above each inlet to prevent potential pipe flooding caused by heavy rain. The equivalent black carbon mass concentrations were obtained from a Multi-Angle Absorption Photometer (MAAP, model 5012, Thermo Electron Group), which is part of the long-term aerosol monitoring program at the ATTO site (Pöhlker et al., 2018; Saturno et al., 2018).

#### 2.2.1 A11 Nano Condensation Nucleus Counter (nCNC)

The nCNC includes a particle size magnifier (PSM, model A10, Airmodus) and a condensation particle counter (CPC, model A20, Airmodus). The PSM acts as a pre-conditioner for the CPC, utilizing diethylene glycol (DEG) as a working fluid to activate and grow nano-sized particles, enabling their detection by the CPC (Vanhanen et al., 2011). Throughout the entire measurement period, the PSM was operated in scanning mode, providing size-resolved information on particles in an approximate size range of 1 to 4 nm with a time resolution of 4 minutes.

The cut-off diameter of the detected particles is dependent upon the mixing ratio of DEG vapors, which in turn is controlled by varying the flow rate of saturated DEG vapors. Therefore, the size distribution can be determined by measuring total particle concentration above a certain cut-off diameter as a function of the flow rate through the saturator (Chan et al., 2020). For this study, we strictly followed the previously established standard operating procedure to ensure accurate instrument configuration, calibration, and measurement performance (Lehtipalo et al., 2022). A straight 20 cm long inlet was used to minimize particle losses within the sampling line. Due to a technical issue in the first phase of our measurement, a design of core sampling (for inlet) was only used during the second phase to further reduce diffusion losses of particles and ensure laminar flow conditions in the sampling line. To better judge the effect of the core sampling, size-dependent transmission efficiency was calculated based on the work of Von Der Weiden et al. (2009). **Figure S1** shows the transmission curves for the scenarios with and without core sampling. Specifically, the transmission efficiencies for 1.5 nm and 3.5 nm particles were 0.62 and 0.85 without the core sampler, and increased to 0.75 and 0.90 with the core sampler, respectively. This highlights the improved performance of core-sampling in reducing particle loss for sub-3 nm particles. The transmission curves were used to correct for the data obtained via the nCNC. The daily routine check was performed to assess the background level of particles potentially caused by DEG homogeneous nucleation. The median background value corresponding to the highest DEG flow rate was 4.7 $cm^{-3}$, which was subtracted from the raw counts for subsequent data analysis.

### 2.2.2 Neutral cluster and Air Ion Spectrometer (NAIS)

The NAIS instrument is a multi-channel aerosol instrument designed to measure the mobility distribution of airborne ions and the size distribution of aerosol particles. It operates over a mobility range of 3.2 to 0.0014 cm²/V/s, corresponding to a mobility diameter range of 0.8 to 40 nm. By simultaneously collecting signals from multiple electrometers, the NAIS can detect both ions and particles. The choice of measuring ions or particles is associated with the operating mode of the preconditioner, which includes ion mode, particle mode and offset mode. By switching the preconditioning unit on and off, the NAIS can achieve a measurement switch between ion mode and particle mode. Moreover, the offset mode is utilized for the periodic verification of the instrument's operation, including the evaluation of noise levels and the measurement of parasitic currents (Manninen et al., 2016). During our field measurements, the NAIS was operated between these three modes, each lasting for 60 seconds. Therefore, an entire measurement cycle took 3 minutes. The NAIS inlet originally provided with the instrument was a 50 cm long metal tube (inner diameter: 35 mm) with a 90-degree downward bend to prevent rain from entering the instrument. A metal mesh (1 mm mesh size) was placed at the front of the inlet tube to keep big insects out. The sample flow rate of the NAIS system was 54 L $min^{-1}$, resulting in a short residence time, reducing diffusion losses and electrical losses for charged clusters and nanoparticles (Manninen et al., 2016). The particle mode of NAIS operates using a corona charger, which can potentially influence the detection of the smallest particles. Moreover, the lower detection limit typically ranging between 2 and 3 nm also depends on corona voltage and ambient gas composition (Manninen et al., 2016). Usually, the results for particles below 2 nm were not used for data analysis.

### 2.2.3 Nano-Scanning Mobility Particle Sizer Spectrometer (SMPS)

The SMPS system consists of an Electrostatic Classifier (model 3082, TSI), a Differential Mobility Analyzer (DMA, model 3086, TSI), a Nano-enhancer (model 3777, TSI) and a Condensation Particle Counter (CPC, model 3772, TSI). Operating at an aerosol flow rate of 2.5 L $min^{-1}$ and a sheath flow rate of 10 L $min^{-1}$, SMPS can achieve number size distribution measurements in the mobility diameter range of 1.3-38.5 nm. During our field measurements, the system was operated without the Nano-enhancer because of its non-functionality. Using an aerosol flow of 1 L $min^{-1}$, the measured size range was adjusted to 10-38.5 nm because the minimum particle size detected by the CPC was 10 nm. The following discussions also include data from another long-term SMPS measurement (particle size range: 40-414 nm), which was carried out at the INSTANT tower located 600 m east of the ATTO tower. The sampling inlet was positioned at a height of 60 m, which was only slightly higher than that of the SMPS on the ATTO tower. More detailed information regarding this long-term SMPS setup can be found in Franco et al. (2022) and Pöhlker et al. (2016).

### 2.2.4 Instrument intercomparison

Prior to the ambient measurements, a laboratory intercomparison was carried out between two systems: PSM A10 + CPC A20 (Airmodus) and the Nano-enhancer 3777 + CPC 3772 (TSI). A similar procedure was also carried out in previous studies (Vanhanen et al., 2011; Kangasluoma et al., 2018; Lehtipalo et al., 2022). The details of the intercomparison experiment setup are illustrated in **Fig. S2**. Tungsten oxide particles with a diameter of 1.4 to 3.8 nm were used, and the desired particle size was achieved through the optimization of wire current and carrier gas flow rate. Subsequently, particles of varying sizes were selected using a DMA, with specific diameter particles

simultaneously measured by both systems. For particles below 1.8 nm, the measurement was conducted without the charger, as the WOx generator (model 7860, GRIMM) produced sufficient self-charged clusters through hot-wire generation (Kangasluoma et al., 2015).

**Figure 1(a)** shows the linear correlation between the concentrations measured by the two systems (Airmodus and TSI) during a laboratory instrument intercomparison prior to the field campaign. The linear fit line with the $R^2$ value of 1.0 indicates a high degree of agreement between the two systems. **Figure 1(b)** illustrates the size-dependent concentration ratio under different sizes. The discrepancy in concentration between the Airmodus instruments and the TSI setup remained below 20% for particle diameters above 1.8 nm, indicating a good agreement between the measured concentrations by the two systems. This level of agreement is consistent with previous intercomparison studies, which reported discrepancies typically within ±20-35% under similar experimental conditions (Cai et al., 2018; Kangasluoma et al., 2015). It is important to note that the effect of charging efficiency may increase the uncertainty of the real ambient measurements due to the different chemical compositions of measured particles (Kangasluoma et al., 2020). In addition to chemical composition, several other factors including different charge states and relative humidity, may also contribute to the overall uncertainty (Lehtipalo et al., 2022).

### 3 Results and discussion

### 3.1 Number size distribution of particles at ATTO

**Figure 2** illustrates the temporal evolution of meteorological parameters and aerosol properties for the entire measurement period. Wind speed was typically low, with an average of 1.8 m s$^{-1}$ over the entire measurement period which is well within the typical range of 0-4 m s$^{-1}$ observed at ATTO (Andreae et al., 2015). The mean temperature was 25.8 ± 3.0 °C and the mean RH was 85.0 ± 13.9 %. The temperatures in December 2022 and January 2023 were 26.9 ± 3.2 °C and 25.0 ± 2.6 °C, and the corresponding RH were 78.8 ± 14.4 % and 89.9 ± 11.6%, respectively. Long-term measurements have shown that temperatures peaked in dry season, with an average of 27.5 °C in September, while the lowest temperature occurred in wet season in March (Andreae et al., 2015). **Figure 2c** displays the combined particle number size distribution derived from NAIS and the ATTO long-term SMPS. We observed several instances of 'Amazonian bananas', characterized by particle growth initiating at a diameter between 20 and 40 nm, consistent with the observations reported by Pöhlker et al. (2018) and Franco et al. (2022). Notable examples include December 7 and January 20, which are highlighted in **Fig. 2c**. In most previous studies in Amazon, the reported lower size limit for the measured particles was generally around 10 nm (Franco et al., 2022; Rizzo et al., 2018; Martin et al., 2010). Here, we further present the number size distribution information of particles below 10 nm. **Figure 2d** shows the total number concentrations in different size ranges. For the whole measurement period, the total particle number concentration ranged from 350 to 4110 cm$^{-3}$ and the median concentration was 969 cm$^{-3}$. There was a large number of particles between 1.5 nm and 3.5 nm, accounting for up to 59 % of the measured total number concentration. It should be noted that, due to their differing measurement principles, the NAIS reports electrical mobility diameters, whereas the PSM measures condensation activation diameters (Kelvin equivalent sizes).

As illustrated in **Fig. 2c**, there were multiple instances of needle-shaped bursts in NAIS measurement during the observation period. The bursts exhibited a good correlation with precipitation events, as evidenced by the RH and rainfall data. **Figure S3** shows that the median values remained within the 330-400 cm$^{-3}$ range across different RH ranges. The high RH range (90-100%) had significantly more outlier values, up to 12000 cm$^{-3}$. Previous studies have also shown that heavy rainfall can cause bursts in the measured ion and particle concentrations by NAIS (Manninen et al., 2016; Hirsikko et al., 2007; Wimmer et al., 2018; Tammet et al., 2009). The proposed explanation is that the splashing of water during heavy rainfall can generate balloelectric intermediate ions, which exist as singly charged water nanoparticles (Tammet et al., 2009). Accordingly, all data potentially affected by precipitation were carefully checked and excluded from the NAIS dataset prior to further analysis. This was done by manually identifying periods with RH = 100% and visually inspecting the particle number size distributions for needle-like burst anomalies typically associated with rainfall events. Additionally, in **Fig. 2c**, a noticeable difference was observed between NAIS and SMPS measurements around the stitching point (40 nm), which has also been reported in previous studies (Yao et al., 2018; Olin et al., 2022). This likely reflects differences in detection principles, instrument uncertainties, and sampling losses near their respective detection limits. SMPS may underestimate small particle concentrations due to diffusional losses and CPC cut-off limitations (Kangasluoma et al., 2020; Von Der Weiden et al., 2009), while NAIS may overestimate them due to high-

sensitivity electrometer detectors and uncertainties in the corona charging process (Mirme et al., 2007; Manninen et al., 2016).

No typical regional NPF events were observed during the entire measurement period, as indicated by the measured particle number size distribution in **Fig 2c**. Typical regional NPF involves a rapid increase in the concentration of sub-3 nm particles in the atmosphere, followed by a continuous growth to larger particle sizes (Kulmala et al., 2012). Previous ground-based measurements have consistently demonstrated the absence of NPF events within the Amazon rainforest PBL (Franco et al., 2022; Wimmer et al., 2018; Rizzo et al., 2018). Although a few NPF events were observed, they were attributed to either the influence of urban pollution plumes (Wimmer et al., 2018), or the initial growth from particles larger than 10 nm (Rizzo et al., 2018; Franco et al., 2022). The growth of particles between 10 and 50 nm in the PBL was found to be closely linked to precipitation events, especially deep convective processes, which served as potential sources to increase the number concentration of nucleation mode particles (Wang et al., 2016; Machado et al., 2021; Andreae et al., 2018; Franco et al., 2022). Machado et al. (2024) analyzed aerosol and trace gas variations during downdrafts and proposed that the observed particle formation was triggered by a sudden increase in ozone levels and a decrease in the condensation sink.

It still remains unclear why there is no typical NPF within the Amazon PBL. On one hand, water vapor may contribute to the suppression of typical NPF events via increasing the condensation sink and scavenging nucleation precursors (Hamed et al., 2011; Andreae et al., 2022). However, the mechanism proposed by Hamed et al. (2011) revealed that the reduced $H_2SO_4$ production was due to lower OH levels at very high relative humidity (RH >80 %), which may not apply under conditions where nucleation is driven primarily by organic vapors. Laboratory studies by Heinritzi et al. (2020) have shown that changes in humidity do not significantly affect the formation rate of highly oxidized molecules (HOMs) for monoterpene- and isoprene-mixed precursor systems. Additionally, recent measurements in free troposphere indicate that NPF can consistently occur under elevated RH conditions, particularly in the outflow and detrainment layers of convective clouds (Xiao et al., 2023). On the other hand, water vapors even play essential role in cluster formation, forming sulfuric acid and stabilizing other species (Stolzenburg et al., 2023). Overall, these findings highlight the complex and still not fully understood role of water vapor in particle formation and growth, which likely depends on the prevailing chemical regime and meteorological conditions. Although typical NPF was absent during our measurement period, a high number concentration of particles below 6 nm was constantly observed (**Fig. 2c** and **2d**). Field measurements in the boreal forest during nighttime have indicated that the clusters formed from HOMs typically do not exceed a few nanometers in diameter (Bianchi et al., 2019). The proposed reason for this is the lack of photochemistry and essential vapors (Rose et al., 2018). These vapors are most likely HOMs resulting from the oxidation of volatile organic compounds (Rose et al., 2018; Ehn et al., 2014). The contribution of HOMs to particle nucleation and growth is strongly dependent on the volatility of HOMs and the particle size, with compounds of lower volatility being more important in the early growth (Tröstl et al., 2016). Other studies have concluded that isoprene can interfere with NPFs associated with monoterpene oxidation, and high levels of isoprene generally inhibit particle formation (Stolzenburg et al., 2023; Lee et al., 2019; Kiendler-Scharr et al., 2009). The involved mechanisms include: (1) scavenging of OH radicals by isoprene , thus reducing the rate of monoterpene oxidation (Mcfiggans et al., 2019); (2) the increased reaction probability between the $RO_2$ radicals generated during α-pinene and isoprene oxidation processes could result in a higher production rate of more volatile $C_{15}$ dimers compared to condensable $C_{20}$ dimers, thus suppressing nucleation and early growth (Heinritzi et al., 2020).

**Table 1** gives an overview of the observed number concentrations of particles and ions across different size ranges for the measurement periods. Based on the particle size classification method proposed by Wimmer et al. (2018), the median concentrations of intermediate (2-4 nm) and large (4-12 nm) particles during the entire measurement period were 749 and 403 $cm^{-3}$, respectively (Table 1). The corresponding concentrations in December were 931 and 497 $cm^{-3}$, much higher than 558 and 306 $cm^{-3}$ observed in January. The study by Wimmer et al. (2018) at another site of the Amazon reported concentrations of 404 and 141 $cm^{-3}$ respectively for the intermediate and large particles during the dry season, and 358 and 115 $cm^{-3}$ during the wet season. In contrast to their ground-based observations, the ATTO measurements were made above the rainforest canopy and therefore are supposed to be more representative of the boundary layer conditions. During wet season, airborne particles tend to be washed out by more frequent rain events (Wimmer et al., 2018; Rizzo et al., 2018).

**Figure 3** compares the number concentrations of particles in different size ranges between December and January, based on PSM measurements. The median total particle number concentrations ($N_{1.5-1000\,nm}$) at the ATTO site were 963 $cm^{-3}$ and 972 $cm^{-3}$ for December and January, respectively. No significant difference was observed between

these two months. In December, the median concentration of particles with diameters from 1.5 to 3.5 nm was 371 cm$^{-3}$; while in January it increased to 573 cm$^{-3}$, indicating a prominent increase in concentration during the pre-wet season compared to the late dry season.

In contrast to particles with diameters below 3.5 nm, particles larger than 3.5 nm exhibited higher median concentrations in December (481 cm$^{-3}$) than in January (335 cm$^{-3}$). Zhou et al. (2002) performed measurements on particles in a similar size range (3-850 nm) during the wet season (March and April) at the Balbina site in the Amazon region using a differential mobility particle sizer (DMPS), and their results yielded a mean concentration around 450 cm$^{-3}$. Further multi-year observations at another site in the Amazon rainforest showed that the median particle number concentration (10-600 nm) was significantly higher in the dry season (1254 cm$^{-3}$) than in the wet season (403 cm$^{-3}$) (Rizzo et al., 2018). Long-term SMPS measurements also revealed that the total number concentrations of particles in the size range of 10-400 nm were, on average, higher in December than in January at the ATTO site (Franco et al., 2022). During the dry season, biomass burning and anthropogenic emissions strongly impact the abundance of larger size particles in the Amazon region (Artaxo et al., 2022).

**Figure 4** shows the abundances of sub-3 nm particles across diverse environments. Here, we use sub-3 nm to denote diameters from 1.5 to 3.5 nm for comparison with previous studies. **Table S1** lists their specific concentrations, size ranges, and site types. Under pristine conditions in the Amazon region, the particle number concentration (491 cm$^{-3}$) was notably lower than in megacities (> 8500 cm$^{-3}$) such as Nanjing, Shanghai and San Pietro Capo Fiume (Xiao et al., 2015; Kontkanen et al., 2017). Sub-3 nm particle number concentration in the Amazon rainforest was also slightly lower compared to those observed in specific rural areas, such as Long Island, USA (590 cm$^{-3}$) (Yu et al., 2014) and Mt. Puy de Dôme mountain, France (500 cm$^{-3}$) (Rose et al., 2015). As for the boreal forest site in Hyytiälä (Kontkanen et al., 2017), its concentrations (580-2900 cm$^{-3}$) are higher than that of the other rural sites, indicating that human activities close to the forest site have an essential impact on the aerosol particle properties(Artaxo et al., 2022).

### 3.2 Number size distribution of naturally charged ions

Naturally charged ions are present in the atmosphere and their average concentrations globally range from 200 to 2500 cm$^{-3}$ depending on the type of measurement sites (Hirsikko et al., 2011). The ion number size distribution obtained from NAIS is shown in **Fig. S4**. According to the classification criteria by Manninen et al. (2016), naturally charged ions can be classified into cluster, intermediate and large ions, with equivalent mobility diameters of 0.8-2, 2-7 and 7-20 nm, respectively. As shown in **Fig. 5**, median concentrations of the cluster ions in December and January were 610 cm$^{-3}$ and 639 cm$^{-3}$, respectively. These numbers are slightly lower than previous measurements in the Amazon rainforest, where the authors reported median concentrations up to 856 cm$^{-3}$ and 952 cm$^{-3}$ in the wet and dry seasons, respectively (Wimmer et al., 2018). In contrast, the median concentrations of intermediate and large ions at the ATTO site remained relatively lower, with their respective concentrations quite comparable, i.e., 40 cm$^{-3}$ versus 35 cm$^{-3}$ for December and 23 cm$^{-3}$ versus 24 cm$^{-3}$ for January. These values, however, are 2-3 times higher than those found by Wimmer et al. (2018). This discrepancy may be related to the different measurement locations/sites. In the previous study, the NAIS was placed inside the canopy in a hut with the sampling inlet 2 meters above the ground (Wimmer et al., 2018). In our study, the NAIS was sampling via the original 50 cm long metal inlet positioned above canopy on the 54-meter-high platform of the ATTO tower.

### 3.3 Diurnal variation patterns of particles and ions

**Figure 6** presents the diurnal variation patterns of the number concentrations of particles of different sizes obtained from the PSM measurements. For both months, the particles within the size range of 1.5-1000 nm showed a clear diurnal trend. After sunrise (around 06:00 local time, LT), the total concentration of particles started to increase, reaching its maximum around 14:00, and then began to decrease steadily until midnight, with the lowest levels occurring around 01:00-03:00. The concentration of particles with diameters above 3.5 nm also increased slightly in the morning in December, and their minimum and maximum values occurred around 5:00 and 17:00, respectively. The latter group of particles, however, showed less pronounced diurnal variation in January.

The particle concentrations in the size range of 1.5-3.5 nm also increased around sunrise for both measurement periods. Their peak concentrations occurred in the morning (08:00 -10:00), and the minimum concentrations were found in the night (around 03:00). The more significant variation in their diurnal pattern was observed for January 2023. This could indicate a sustained growth over a longer period for smaller particles under cleaner conditions. The study at a pasture site in the Amazon during the dry season showed an increase in the median concentrations

of sub 3 nm particles before sunrise (03:00-06:00), followed by a decrease in the early afternoon (12:00-15:00) and finally an increase in the evening (18:00-24:00) (Wimmer et al., 2018). The authors attributed these patterns to a so-called Carnegie curve, which reflects the diurnal variation pattern of the ionospheric potential (Wimmer et al., 2018). Other studies have also reported daytime maxima in the concentrations of particles smaller than 3 nm, which were commonly associated with NPF events (Kulmala et al., 2013; Yu et al., 2014; Sulo et al., 2021). For the daytime maximum of sub-3 nm particle concentrations observed on non-event days, it was likely associated with the photochemical processing of low volatile gas precursors (Kontkanen et al., 2017).

**Figure. 7** illustrates the daily variation in the size distribution of particles within the 2-40 nm range measured by NAIS. To better illustrate the dynamic changes in particle number concentrations, we separately analyzed particles in the 2-6 nm, 6-20 nm, and 20-40 nm size ranges. Multimodal particle size distributions that correspond to these sizes have been reported in previous studies (Rissler et al., 2004; Zhou et al., 2002). The number concentration for 2-6 nm particles steadily increased throughout the daytime, peaking in the late afternoon for both December and January (see also in **Figure S5**). Rissler et al. (2004) also observed a pronounced nucleation mode around 16:00-18:00 local time, followed by a decrease and a secondary rise around 06:00-07:00 in the early morning. In December, the concentration of 6-20 nm particles also increased notably from 08:00 in the morning to the afternoon, shortly after the initial rise of 2-6 nm particles. The phenomenon of lagged increasing suggests particle growth processes. In January, particles in the 20-40 nm size range exhibited a continuous increase during nighttime (18:00-24:00), reaching a maximum in the early morning hours. This behavior, however, was not observed in December. Meanwhile, a comparable diurnal variation pattern was also identified for the number concentrations of sub-50 nm particles at the same site, based on long-term measurements conducted over more than six years (Franco et al., 2022). The authors attributed the observed decrease in number concentration in the late morning to the increased mixing following the destruction of the nocturnal boundary layer and the subsequent formation of a well-mixed boundary layer (Franco et al., 2022).

**Figure. 8** displays the diurnal patterns of naturally charged ions within different size ranges. The concentration of cluster ions showed a similar daily pattern in both months, with the highest concentrations observed in the morning (~08:00 - 11:00 LT). Throughout the day, the concentrations were relatively stable with minor fluctuations. The overall range of cluster ion concentration was between 500 cm$^{-3}$ and 700 cm$^{-3}$ and was constantly higher than intermediate and large ions. The concentrations of intermediate and large ions increased slightly during the day and remained at relatively constant levels throughout the night, with their maximum concentrations appeared in the afternoon.

**3.4 Comparisons of diurnal variation patterns between polluted and clean days**

Black carbon (BC) can serve as a unique marker to estimate and track the potential contributions from biomass burning activities in the Amazon Basin and long-term transport of African smoke (Pöhlker et al., 2018; Holanda et al., 2023). To assess the background particle levels in the pristine environment, we divided the entire observation period into two scenarios: polluted days affected by short- and long-distance transported plumes, and clean days with extremely clean conditions. Following the approach described in Pöhlker et al. (2018), a threshold of 0.01 μg m$^{-3}$ for the mass concentration of BC was used to distinguish pristine and plume-affected conditions. Accordingly, a day was considered pristine if the BC mass concentration was continuously below 0.01 μg m$^{-3}$ for more than 6 hours; otherwise, it was categorized as a polluted day.

The measurement period was divided into 38 polluted days and 7 clean days. Only 18 % of the total measurement days were truly pollution-free, with two days in December and five days in January. The median concentration of particles below 3.5 nm were 441 and 1153 cm$^{-3}$ on polluted and clean days, respectively. However, particles larger than 3.5 nm were more abundant on polluted days, with a median concentration of 388 cm$^{-3}$, compared to 315 cm$^{-3}$ on clean days. The higher concentrations of large particles on polluted days could be related to biomass burning or long-term transport. On the other hand, these large particles provided more aerosol surfaces to act as a sink for condensable gas precursors (condensation) and nanoparticles (coagulation), resulting in lower concentrations of sub-3 nm particles on polluted days.

The diurnal patterns of the number concentrations of particles measured by PSM are presented in **Fig. 9**. The sub-3 nm particles exhibited much more pronounced variations on clean days compared to polluted days. During clean days, the number concentration of sub-3 nm particles increased significantly in the morning, reaching a peak value around 12:00 LT, and then steadily decreased until midnight. For particles with diameters between 3.5 nm and 1000 nm, their concentrations were rather flat on both clean days and polluted days. The number concentration

of sub-3 nm particles was significantly higher during the daytime on clean days. This indicates that photochemistry has a significant impact on the production of particles below 3 nm. However, the number concentration of sub-3 nm particles remained relatively low compared to typical NPF events (Hong et al., 2023; Deng et al., 2022; Bianchi et al., 2016; Kulmala et al., 2013). The absence of clear growth suggests either a lack of condensable vapors to sustain particle growth or efficient loss processes such as coagulation with larger particles (Stolzenburg et al., 2023). During the night, the difference between the two scenarios was less pronounced, although the concentration of sub-3 nm particles on clean days was still slightly higher than that on polluted days. Previous observation in Hyytiälä reported high concentrations of sub-3 nm particle in the evening, and this phenomenon has been proposed to be related to ozonolysis products of monoterpenes (Lehtipalo et al., 2009).

**Figure 10** illustrates the diurnal patterns of the particle size distributions on clean and polluted days. As shown in the left panel of **Fig. 10**, the number of nucleation mode particles is higher on clean days. Specifically, the particle concentrations in the 2-6 nm range started to increase around 08:00 (see also in **Figure S6**), and 6-20 nm particles showed growth and elevated concentrations in the afternoon. The diurnal variation observed during polluted days (the right panel of **Fig. 10**) exhibited a pattern similar to that described in **Fig. 7**. It can be inferred that the more pronounced increase of nucleation mode particles is linked to the low coagulation sinks during clean days. Nevertheless, the potential formation mechanisms of these nucleation particles remain to be elucidated through further investigation.

**4 Conclusion**

The Amazon rainforest provides a unique pristine environment that offers the opportunity to investigate atmospheric aerosol particles during episodes that are free of anthropogenic pollution. In this study, state-of-the-art aerosol instruments including PSM, NAIS, and SMPS were deployed at the ATTO site for an intensive field measurement as part of the "CAFE-Brazil" campaign from December 2022 to January 2023. The size distributions of particles and ions smaller than 40 nm were obtained. During these two months of measurements, a large number of sub-3 nm particles and cluster ions were constantly observed in the Amazonian PBL. Based on the PSM measurements, we found that the median concentrations of particles below 3.5 nm were 371 and 573 $cm^{-3}$ in December and January, respectively. These values represented 38% and 59% of the total number concentrations of particles in the size range of 1.5 nm to 1 μm. The NAIS measurements revealed that the median concentration of naturally charged ions in the range of 0.8-40 nm (equivalent mobility diameter) was 761 $cm^{-3}$. In addition, the median concentrations of cluster (0.8-2 nm), intermediate (2-7 nm) and large (7-20 nm) ions were 624, 33, and 30 $cm^{-3}$, respectively. NAIS also measured the median concentration of particles with diameters between 2 and 40 nm as 1462 $cm^{-3}$. These results highlight the significant contribution of nucleation mode particles to the total particle number concentration in the Amazon rainforest.

Although a large number of sub-3 nm particles and naturally charged ions were present in the Amazonia boundary layer, typical NPF events were not observed throughout the entire measurement period. However, the particles and ions below 3 nm showed a clear diurnal variation. Their concentrations started to increase around sunrise, peaked around midday, and then slowly decreased until midnight. This pattern is an indication that the photochemistry that converts the biogenic VOCs into low volatility vapors may be responsible for the formation of these nanoclusters. In addition, boundary layer convective mixing and rainfall events might also contribute to the diurnal variation to some extent. Particles with diameters larger than 3.5 nm and intermediate/large ions only displayed small diurnal variations. Under pristine conditions, we observed higher number concentrations and more pronounced diurnal patterns for sub-3 nm particles and ions. However, when influenced by biomass burning and other transported plumes, their diurnal patterns were significantly suppressed.

This study is the first to provide a detailed description of the size distribution and diurnal variation of particles and ions smaller than 3 nm measured above the forest canopy in the central Amazon region, supposed to offer a more accurate representation of boundary layer conditions of the region. In future work, we aim to elucidate whether these clusters in the PBL of the Amazon rainforest originate from biogenic VOC precursors, by integrating the gas-phase measurement data obtained from chemical ionization mass spectrometry (CIMS). By investigating the potential links between those gas species that favor NPF (e.g., HOMs and sulfuric acid) and the nanoclusters, we will further unravel the underlying mechanisms that drive the appearance and/or initial growth of nanoparticles, as well as the reasons why their subsequent growth could not be sustained. All this information will contribute to gain a more comprehensive understanding of the air chemistry in this particular pristine environment, and also provide insights into the atmospheric processes in the areas where the influences of human activities is most pronounced.

 **Tables and Figures**

**Table 1.** A summary of the particle number concentration for each instrument in different size ranges. Numbers represent median values, with the 25th-75th percentiles in parentheses.

| Instrument | Size range (nm) | Number concentration ($cm^{-3}$) | | |
|---|---|---|---|---|
| | | 2022.12 | 2023.01 | All period |
| PSM* | 1.5-1000 | 963 (679-1363) | 972 (775-1287) | 969 (735-1320) |
| | 1.5-3.5 | 371 (251-740) | 573 (406-909) | 491 (305-851) |
| | 3.5-1000 | 481 (355-611) | 335 (258-410) | 380 (295-539) |
| NAIS (particle) | 2-40 | 1799 (1427-2195) | 1141 (883-1489) | 1462 (1076-1944) |
| | 2-4 | 931 (715-1177) | 558 (393-778) | 749 (509-1032) |
| | 4-12 | 497 (400-614) | 306 (233-401) | 403 (290-538) |
| NAIS (Ion) | 0.8-40 | 769 (677-876) | 751 (644-866) | 761 (661-872) |
| | 0.8-2 | 610 (532-712) | 639 (543-746) | 624 (537-728) |
| | 2-7 | 40 (34-50) | 23 (18-31) | 33 (23-43) |
| | 7-20 | 35 (26-45) | 24 (16-35) | 30 (20-41) |
| SMPS | 10-40 | 17 (9-47) | 35 (18-73) | 27 (13-63) |

* PSM measures the condensation activation diameters (Kelvin equivalent), while the sizes reported by other instruments represent electrical mobility diameters.

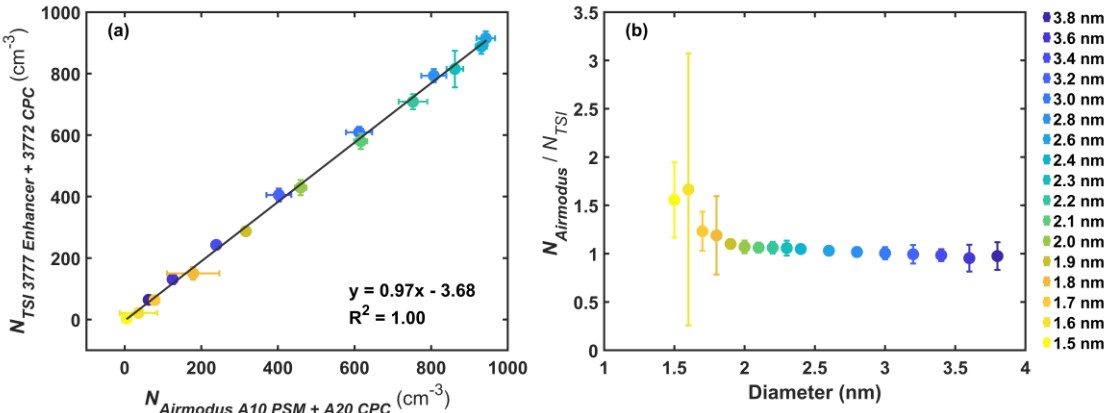

**Figure 1.** (a) Comparison between the median particle number concentrations measured by two systems, namely the Airmodus (A10 PSM + A20 CPC) and the TSI (Nano-enhancer 3777 + CPC 3772) systems, for the selected different sizes (diameters ranging from 1.5 to 3.8 nm). The solid line represents the linear fitting result with an $R^2$ value of 1. (b) The ratio of the particle concentration measured by the two systems at different sizes. All error bars denote the standard error.

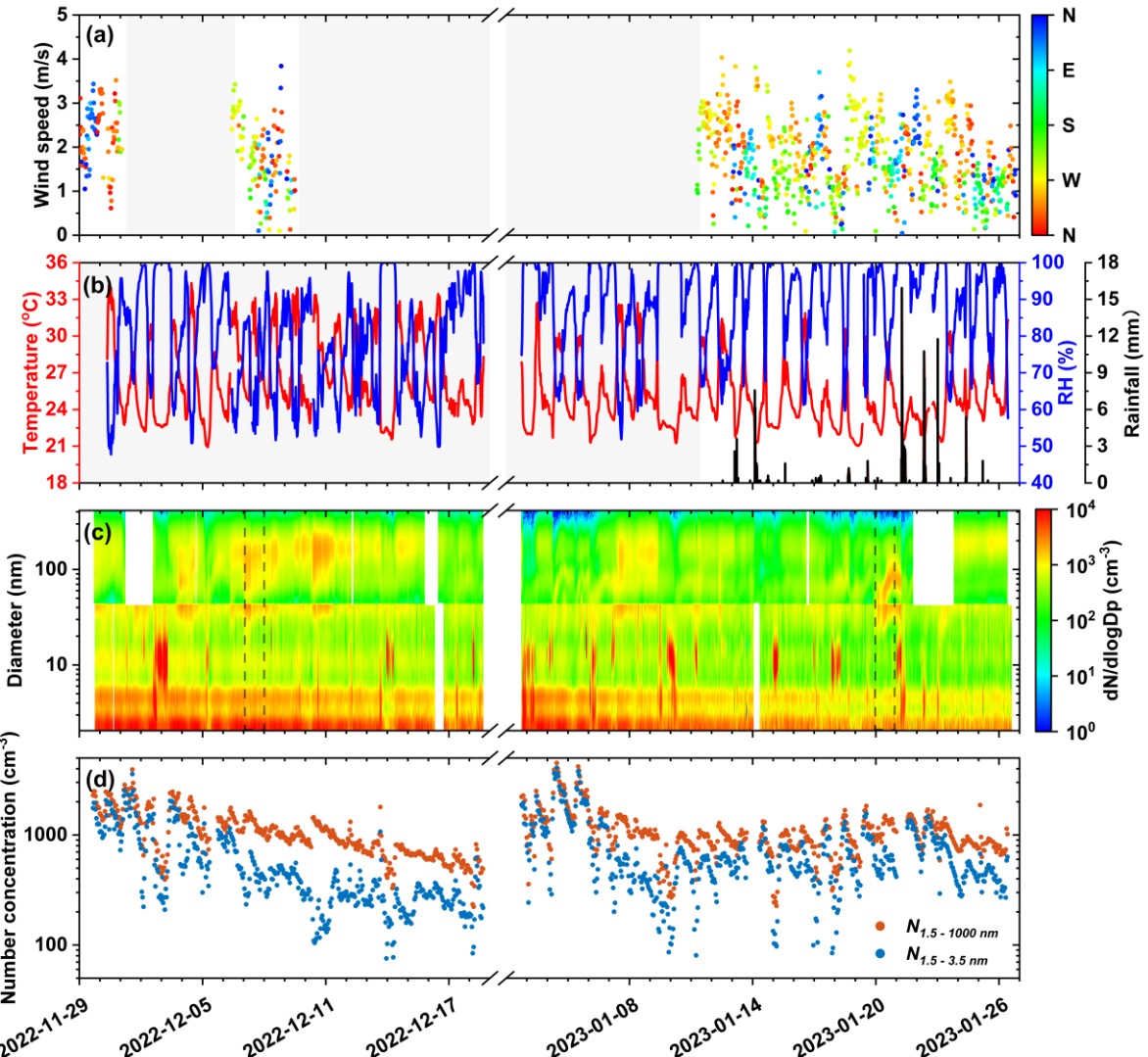

**Figure 2.** The time series plot of (a) Wind speed and wind direction; (b) Ambient relative humidity (RH), temperature, and rainfall; (c) Particle number size distribution from the NAIS (2-40 nm, negative particle mode) and the SMPS system (40-414 nm); (d) Total number concentration of particles with diameter ranges of 1.5-1000 nm and 1.5-3.5 nm from the PSM measurement. Note that the grey parts in panels (a) and (b) and the white part in panel (c) indicate that no data are available. The two dashed rectangles in panel (c) highlight representative 'Amazonian banana' events observed on December 7 and January 20. The timestamps used here are synchronized with the local time in Manaus, called Local Time (LT), which is equal to Coordinated Universal Time (UTC) minus 4 hours.

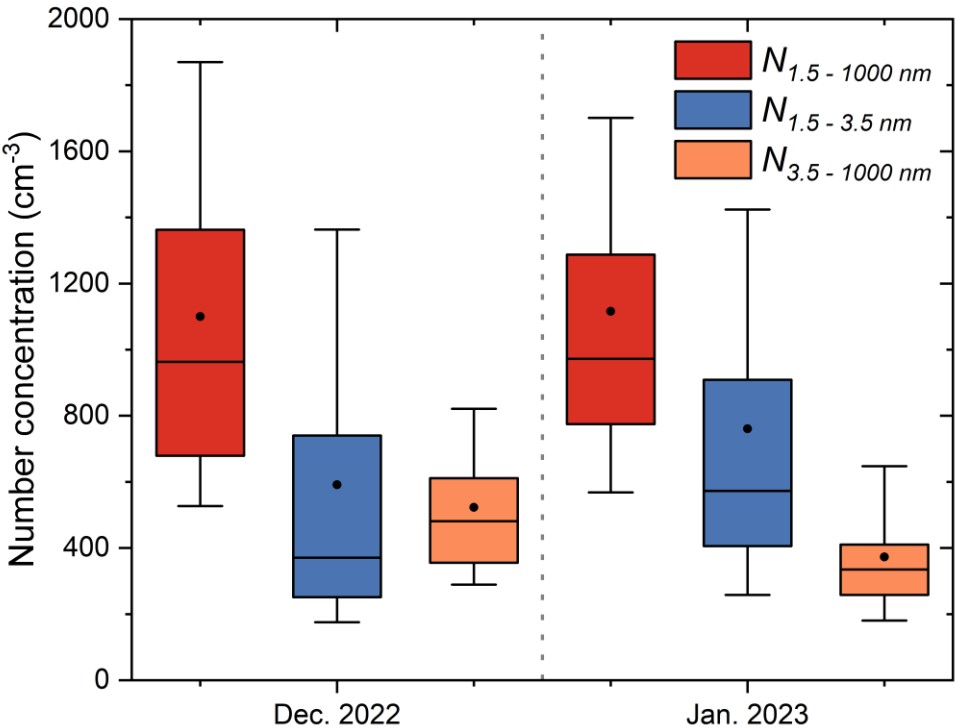

**Figure 3.** Comparison of particle number concentrations in December 2022 and January 2023 for different size ranges based on PSM measurements. The lower/upper boundaries of the box and the vertical whiskers indicate the 25th/75th and 10th/90th percentiles, respectively. Horizontal lines and black dots within the boxes indicate median and mean values, respectively.

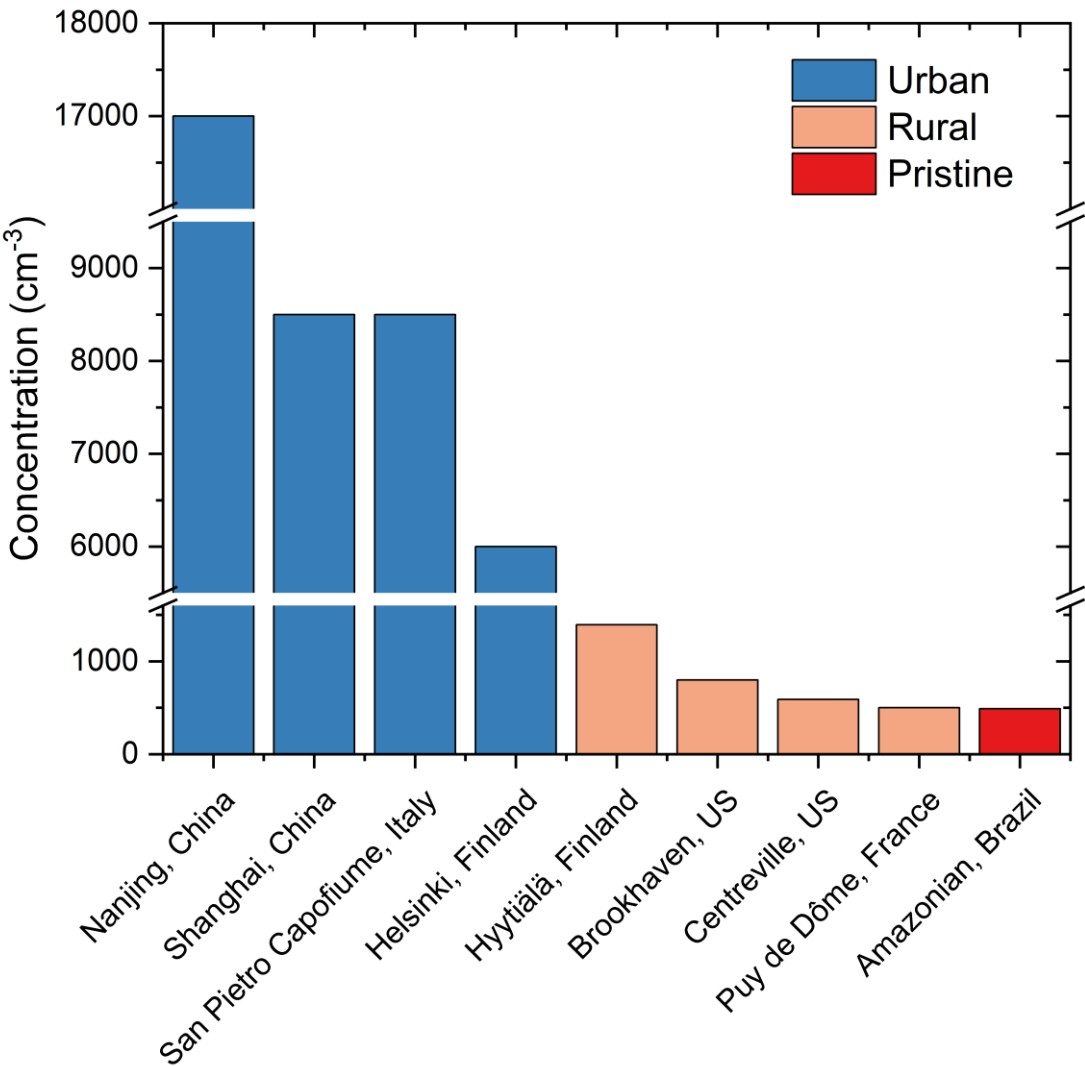

**Figure 4.** Summary of PSM measurements of particle number concentrations below 3 nm in different environments around the globe. All data are from previous studies, except for the results from the Amazonian region, see Table S1 for details.

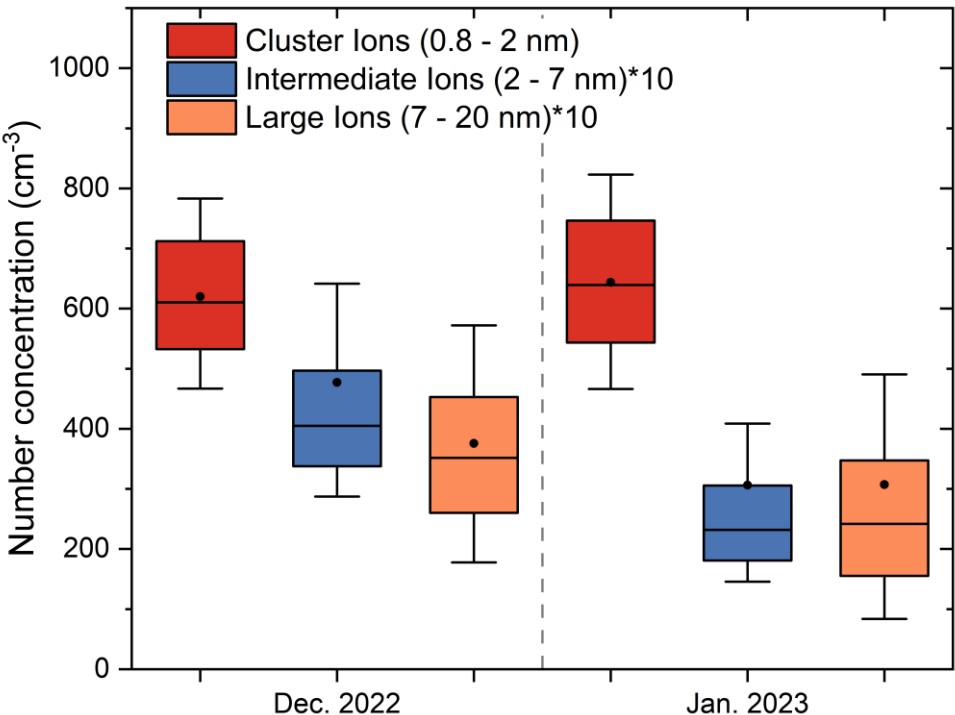

**Figure 5.** Comparison of negative ion number concentrations in December 2022 and January 2023 for different size ranges based on NAIS measurements. The 25th and 75th percentiles are indicated by the bottom and top edges of the box, while the 5th and 95th percentiles are indicated by vertical whiskers. The horizontal line in the box indicates the median value and the black dot indicates the mean value.

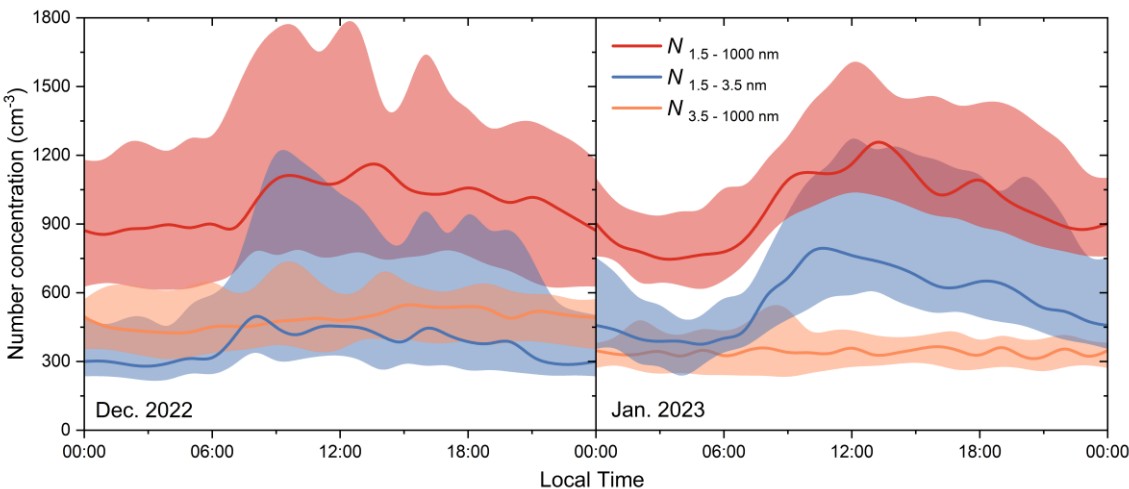

**Figure 6.** Diurnal variation of particle number concentrations in December 2022 (left panel) and January 2023 (right panel), based on PSM measurements of the size ranges of 1.5-1000 nm (in red line), 1.5-3.5 nm (in blue line), and 3.5-1000 nm (in orange line). The left panel shows the December 2022 results and the right panel shows the January 2023 results. Solid lines indicate median values and shaded areas indicate the 25th-75th percentiles.

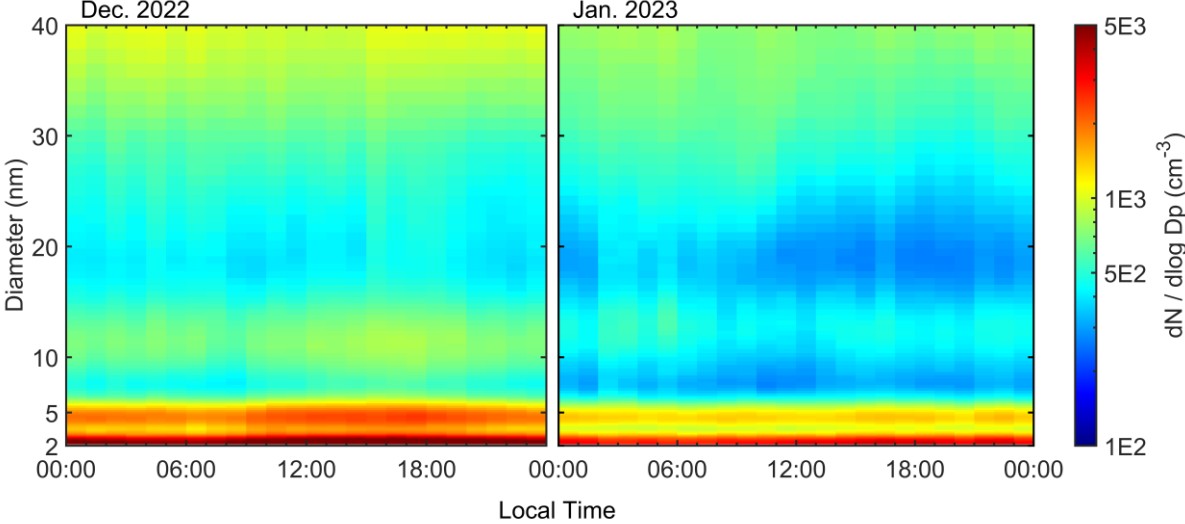

**Figure 7.** Diurnal variation of the particle number size distribution measured by NAIS during December 2022 (left panel) and January 2023 (right panel). The plot data represent the median values.

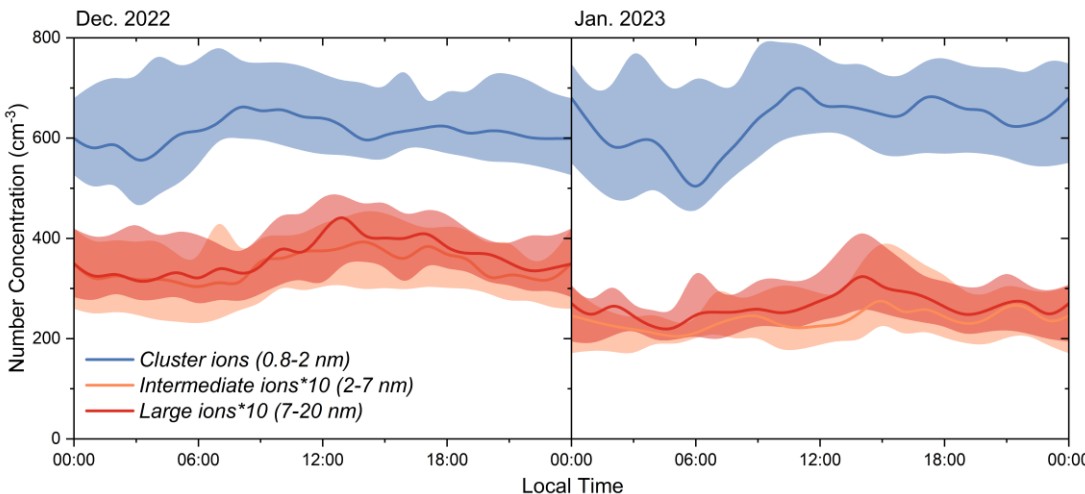

**Figure 8.** Diurnal patterns of cluster ions in the 0.8-2 nm size range (blue), medium ions in the 2-7 nm range (orange), and large ions in the 7-20 nm range (red) based on NAIS measurements. The left panel indicates results for December 2022 and the right panel indicates results for January 2023. The solid lines indicate the median values and the shaded areas indicate the 25th-75th percentile.

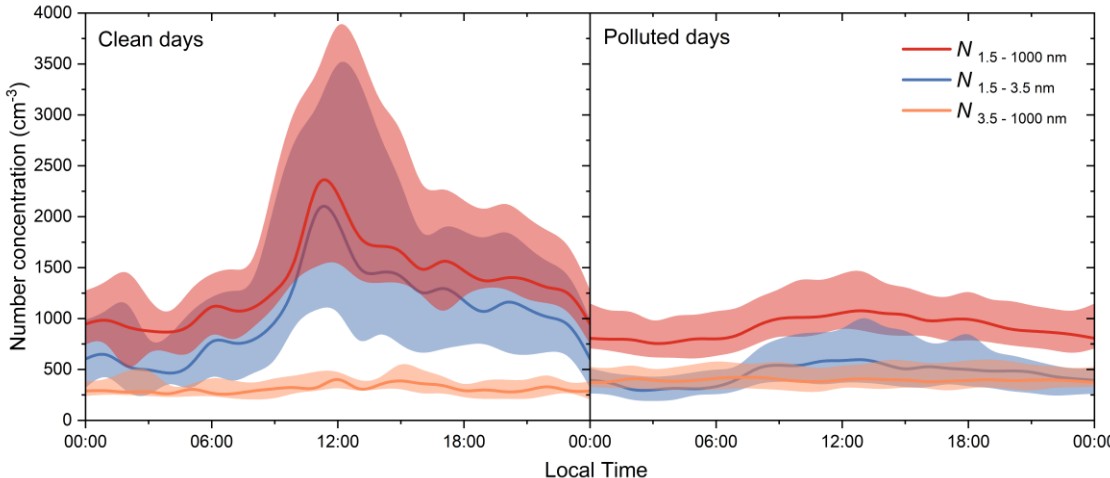

**Figure 9**. Diurnal variation in number concentrations on clean (left panel) and polluted days (right panel) based on PSM measurements for particles with diameters between 1.5 and 1000 nm (red), 1.5 and 3.5 nm (blue) and 3.5 and 1000 nm (orange). The Lines indicate median values and the shaded areas indicate the 25th-75th percentiles.

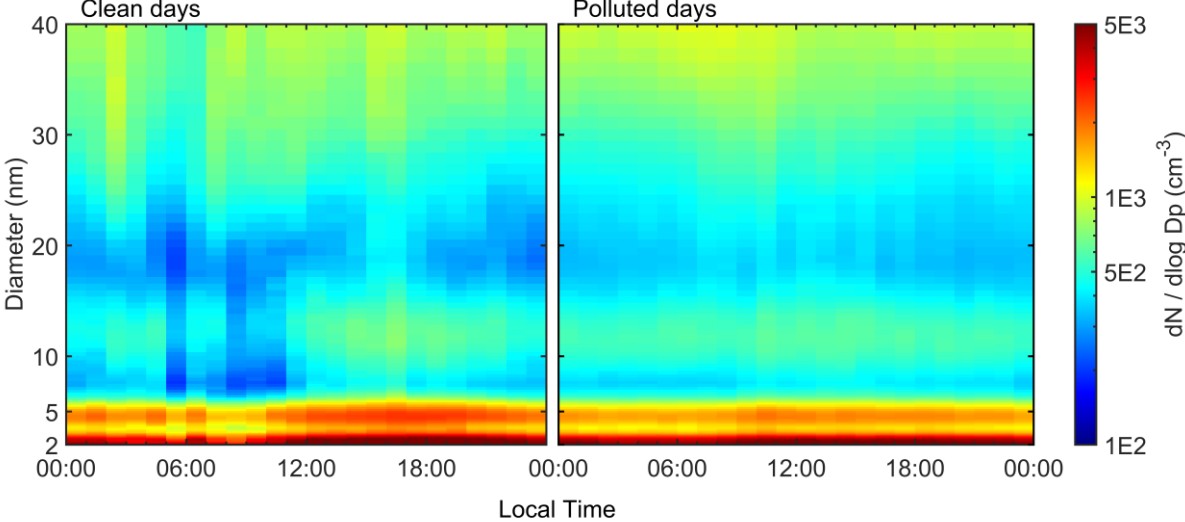

**Figure 10**. Diurnal variation of the particle number size distribution measured by NAIS during clean days (left panel) and polluted days (right panel). The plot data represents the median values.

**Data availability.** The data presented in this work are available at https://doi.org/10.17617/3.ZBHLIR (Jianqiang et al., 2025).

**Author contributions:** HS designed and led the study. JZ, GL, UK, and BM made measurements. JZ collected and processed the data. JZ, GL, and HS interpreted the results. JZ and GL wrote the paper. UK, BM, CP, PA, UP, and YY provided valuable comments and suggestions on the analysis and the manuscript. CP, PA, UP, YC, and HS coordinated the CAFE-Brazil at the ATTO site. All authors participated to the discussion of the results as well as the finalization of the paper.

**Competing interests:** The authors declare that they have no conflict of interest. Some authors are members of the editorial board of the journal Atmospheric Chemistry and Physics.

**Acknowledgements:** For the operation of the ATTO site, we acknowledge the support by the Instituto Nacional de Pesquisas da Amazônia (INPA), the Amazon State University (UEA), the Large-Scale Biosphere-Atmosphere Experiment (LBA), FAPEAM, the Reserva de Desenvolvimento Sustentável do Uatumã (SDS/CEUC/RDS-Uatumã), and the Max Planck Society. Particularly, we would like to thank all colleagues involved in the technical, logistical, and scientific support of the ATTO project. We acknowledge the support by the University of São Paulo (USP) and the Instituto Nacional de Pesquisas da Amazônia (INPA). We would like to thank Reiner Ditz, Thomas Kenntner, Thomas Klimach, Sebastian Brill, Leslie Kremper, Amauri Rodriguês Perreira, Antonio Huxley Melo Nascimento, Rosaria Rodrigues, Cléo Quaresma for technical, logistical, and scientific support within the ATTO project. B.M acknowledges Fundação de Amparo à Pesquisa do Estado de São Paulo (FAPESP) for the scholarships nº 2020/15405-0 and 2023/01902-0.

**Financial support:** H.S. is supported by the Strategy Priority Research Program of Chinese Academy of Sciences (XdB0760000), Chinese Academy of Sciences President's international Fellowship initiative (2024PG0014), and the national Key Scientific and technological infrastructure Project "earth System Science numerical Simulator Facility". The ATTO project is supported by the Max Planck Society (MPG), the Bundesministerium für Bildung und Forschung (01LB1001A, 01LK1602B, and 01LK2101B), and the Brazilian Ministério da Ciência, Tecnologia e Inovação (01.11.01248.00).

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
