# Peer review of "Measurement report: Number size distribution of sub-40 nm particles in the Amazon rainforest"

_EGUsphere, 2024_

## Author Comment (AC1)

**Response to Reviewer #1**

We sincerely thank the reviewer for the valuable feedback and comments, which have helped to improve our manuscript. In the following, we address the general and specific comments point by point. The reviewer's comments are given in ***black bold italic***, while our responses are in normal font (non-bold, non-italic). Revised sections in the manuscript text in response to the comments are written in blue. The page and line numbers refer to the revised version. The figures and tables used in the responses are labeled the same as the revised version.

***Comments and suggestions:***

***In this measurement report, the number size distribution of sub-40 nm particles derived from different instruments in the Amazon rainforest has been reported, including the diurnal pattern and the sources of nanoparticles, especially for the particles below 10 nm. This study provides valuable information in understanding the particle sources in Amazon region and highlight the necessity about the study on nanoparticles. However, further discussion needs to be added to make the results more robust. I recommend this paper being accepted after the below issues are addressed.***

**Responses and Revisions:**

We thank the reviewer for the positive evaluation of our manuscript. We also appreciate the constructive suggestions provided, which have helped us to improve the clarity and robustness of the manuscript. In the revised version, we have carefully addressed all specific comments and incorporated necessary clarifications and additional discussions as recommended.

**Comments and suggestions:**

***Line 45, It is necessary to clearly state that with or without a core sampling device, how much the sampling efficiency was changed for PSM.***

**Responses and Revisions:**

Thanks for the comments. In addition to minimizing the inlet line length to reduce particle loss, we also employed the core sampling device to further improve transmission efficiency for sub-3 nm particles. Figure S1 presents the detailed size-dependent transmission efficiencies for inlet systems with and without core-sampling. Specifically, the transmission efficiencies for 1.5 nm and 3.5 nm particles without the core sampler were 0.62 and 0.85 respectively, whereas with the core sampler, the efficiencies increased to 0.75 and 0.90. These values highlight the improved performance of the inlet system with the core sampler for detecting sub-3 nm particles.

The revised manuscript has included the following clarification sentences (Line 149-151).

"Specifically, the transmission efficiencies for 1.5 nm and 3.5 nm particles were 0.62 and 0.85 without the core sampler, and increased to 0.75 and 0.90 with the core sampler, respectively. This highlights the improved performance of core-sampling in reducing particle loss for sub-3 nm particles."

The sentence also had been added in the revised Supplementary Material (Line 16-17).

"Figure S1. The transmission efficiency for inlet systems with and without core-sampling. The size-dependent transmission efficiency was calculated with the way introduced by Von der Weiden et al. (2009)."

**Comments and suggestions:**

*Section 2.2.4, is there any calibration was conducted before or after for the intercomparison between nCNC and TSI SMPS? Or the two instruments' inherent system error was lower than 20%? It would be better if the authors can the information about if there are any other studies did such intercomparison, otherwise, how could we know the 20% discrepancy is a "good agreement". In figure 1, please make sure the slope is positive or negative.*

**Responses and Revisions:**

Thanks for the valuable comments. During the intercomparison experiments, we also employed a Grimm Faraday Cup Electrometer (Model 5705, shown in Figure S2) to calibrate and determine the actual cut-off sizes for each instrument. The measured cut-off diameters were 1.7 nm for the TSI system and 1.5 nm for the Airmodus PSM. Given that the resolution of the DMA 3086 is not ideal for such small particle sizes, and considering the overall good agreement observed between the TSI and Airmodus systems, we did not apply further adjustments to the calibration curves or settings of the two instruments.

Regarding the reported ~20% discrepancy, we have summarized relevant intercomparison results from previous studies in Table 1. Similar lab experiments using $WO_x$ particles have shown discrepancies within ±20-35%, and even larger deviations (up to a factor of 10) have been reported in chamber experiments. Thus, we consider the results observed in our intercomparison experiments to be within an acceptable and expected range.

Table 1 Summary of the instrument comparisons (Kangasluoma et al., 2020)

| Instruments | Test aerosol | Concentration Range (cm-3) dN/dlogDp | Size range (nm) | Factor between the instruments | Reference |
|---|---|---|---|---|---|
| PSM, SMPS | $WO_3$ | $10^4$-$10^6$ | 1.2-2.6 | <1.2 | (Kangasluoma et al., 2015) |
| PSM, SMPS | $WO_3$ | $10^3$-$10^5$ | 1.5-4 | 1-1.35 | (Cai et al., 2018) |
| PSM, NAIS, SMPS | Chamber experiment | $10^1$-$10^5$ | 1.3-1000 | 1-10 | (Kangasluoma et al., 2020) |
| PSM, SMPS | $WO_3$ | $10^2$-$10^5$ | 1.8-3.8 | 1-1.2 | This study |

We have updated the manuscript accordingly (Line 201-203).

"This level of agreement is consistent with previous intercomparison studies, which reported discrepancies typically within ±20-35% under similar experimental conditions (Cai et al., 2018; Kangasluoma et al., 2015)."

Additionally, we ensured the slope in Figure 1 is clearly indicated and labeled with its correct sign.

[Figure]

Figure 1. (a) Comparison between the median particle number concentrations measured by two systems, namely the Airmodus (A10 PSM + A20 CPC) and the TSI (Nano-enhancer 3777 + CPC 3772) systems, for the selected different sizes (diameters ranging from 1.5 to 3.8 nm). The solid line represents the linear fitting result with an R2 value of 1. (b) The ratio of the particle concentration measured by the two systems at different sizes. All error bars denote the standard error.

**Comments and suggestions:**

*Line 210, it should be better to give the definition of "Amazonian bananas" here, otherwise, it may lead the reader to mistakenly think it refers to banana shaped NPF.*

**Responses and Revisions:**

Thanks for pointing this out. We agree that the term "Amazonian bananas" could be misinterpreted as referring to "banana-shaped" new particle formation (NPF) events. However, the term "Amazonian bananas" does not have a formal definition. Unlike typical NPF events, which begin with particle growth from a few nanometers, the "Amazonian bananas" describe growth patterns that typically start at larger sizes, around 20 to 40 nm, as observed in previous studies (Pöhlker et al., 2018; Franco et al., 2022). We have updated the manuscript accordingly (Line 218-220).

"We observed several instances of 'Amazonian bananas', characterized by particle growth initiating at a diameter between 20 and 40 nm, consistent with the observations reported by Pöhlker et al. (2018) and Franco et al. (2022). Notable examples include December 7 and January 20, which are highlighted in **Fig. 2c**."

**Comments and suggestions:**

*Line 220-227, is the "high RH range (90-100%)" corresponding to the precipitation? How much is the data excluded from the dataset due to precipitation or high RH? As the mean RH during this measurement was ~85%, that means the probability of the RH exceeding 85% is high.*

**Responses and Revisions:**

The high RH range (90-100%) does not always correspond directly to precipitation events, but it often reflects periods with elevated moisture levels, including pre- and post-rain conditions or fog. However, RH values reaching 100% are typically associated with active precipitation.

In addition to monitoring RH, we also inspected the particle number size distributions for characteristic needle-shaped bursts that are related to raining events. In our analysis, we manually excluded the data only during periods with RH = 100% and/or by visual inspection of the particle spectra when they revealed clear anomalies. Similar to the approach by Wimmer et al. (2018), non-reliable data were identified visually using surface plots of particle size distributions. In total, approximately 22 % of the data, mainly from periods with rain or extremely high relative humidity (RH = 100%), were excluded from further analysis to ensure data quality.

Meanwhile, based on six years of measurements at the ATTO site, the RH in the Amazon typically ranges from 75% to 95% over the course of a day (Franco et al., 2022).

We further clarify this point in the manuscript (Line 236-239).

"Accordingly, all data potentially affected by precipitation were carefully checked and excluded from the NAIS dataset prior to further analysis. This was done by manually identifying periods with RH = 100% and visually inspecting the particle number size distributions for needle-like burst anomalies typically associated with rainfall events."

**Comments and suggestions:**

*Line 240, have you looked in to the information about radiation and cloud cover in this study? In addition, as the authors have mentioned the high RH during the measurement, it can also be a reason why no typical NPF was observed. As water vapor can also contribute to the high condensation sink in the ambient air, whereas the CS is normally calculated based on the dry PNSD.*

**Responses and Revisions:**

We thank the reviewer's good suggestion. We agree that factors such as solar radiation, cloud cover, and high relative humidity (RH) can play important roles in impacting new particle formation (NPF) processes. In this study, although we did not include direct measurements of solar radiation or cloud cover, we acknowledge their potential impacts on photochemistry and particle formation. As in Fig. 6 and Fig. 8, we show clearly the sub-3 nm and cluster ions concentration increased as sun arise and peaked around noon, which followed the same trend as the radiation diurnal cycle (Franco et al., 2022). More detailed mechanistic discussions related to photochemistry-induced NPF are out of the scope of this measurement report.

Moreover, we agree that water vapor could potentially contribute to the suppression of typical NPF events by increasing the condensation sink and scavenging nucleation precursors (Andreae et al., 2022; Hamed et al., 2011). However, the mechanism proposed by Hamed et al. (2011) revealed that the reduced $H_2SO_4$ production was due to lower OH levels at very high relative humilities (>80 %), which may not apply under conditions where nucleation is driven primarily by organic vapors. Laboratory studies by Heinritzi et al. (2020) have shown that changes in humidity do not significantly affect the formation rate of highly oxidized molecules (HOMs) for monoterpene- and isoprene-mixed precursor systems. Additionally, recent measurements in free troposphere indicate that NPF can consistently occur under elevated RH conditions, particularly in the outflow and detrainment layers of convective clouds (Xiao et al., 2023). On the other hand, water vapors even play essential role in cluster formation, forming sulfuric acid and stabilizing other species (Stolzenburg et al., 2023). Overall, these findings highlight the complex and context-dependent nature of water vapor's influence on nucleation, underscoring the need for more detailed investigations into its role in both particle formation and growth.

We have added the following sentence into the manuscript (Line 259-271).

"On one hand, water vapor may contribute to the suppression of typical NPF events via increasing the condensation sink and scavenging nucleation precursors (Hamed et al., 2011; Andreae et al., 2022). However, the mechanism proposed by Hamed et al. (2011) revealed that the reduced $H_2SO_4$ production was due to lower OH levels at very high relative humidity (RH >80 %), which may not apply under conditions where nucleation is driven primarily by organic vapors. Laboratory studies by Heinritzi et al. (2020) have shown that changes in humidity do not significantly affect the formation rate of highly oxidized molecules (HOMs) for monoterpene- and isoprene-mixed precursor systems. Additionally, recent measurements in free troposphere indicate that NPF can consistently occur under elevated RH conditions, particularly in the outflow and detrainment layers of convective clouds (Xiao et al., 2023). On the other hand, water vapors even play essential role in cluster formation, forming sulfuric acid and stabilizing other species (Stolzenburg et al., 2023). Overall, these findings highlight the complex and still not fully understood role of water vapor in particle formation and growth, which likely depends on the prevailing chemical regime and meteorological conditions."

**Comments and suggestions:**

**Line 263-264, is that reasonable that the December data can represent the dry season as the mean RH was approximate 79%? The authors may refer to the previous literature to check the typical RH levels during the dry season in Amazon.**

**Responses and Revisions:**

Thanks for raising this important point. According to long-term measurements at the ATTO site spanning from February 2014 to September 2022, relative humidity (RH) typically ranges from approximately 75-95% throughout the year and 80-100% during the wet season (Franco et al., 2022). Wimmer et al. (2018) reported that the median RH inside the rainforest was 96.9% during the wet season and 94.4% during the dry season. During our measurement period, the average RH was $78.8 \pm 14.4\%$ in December and $89.9 \pm 11.6\%$ in January, indicating that our December RH values were lower than the typical dry season values reported in earlier studies. This indicates that our campaign in December experienced much drier conditions.

Regarding the classification of wet and dry seasons, the studies by Pöhlker et al. (2016) and Andreae et al. (2015) defined the dry season as from August to November and the wet season from February to May. However, Wimmer et al. (2018) followed the definition of Artaxo et al. (2013), in which the dry season extends from July to December, and the wet season from January to June. Thus, we generally think that December and January correspond to the end of the dry season and the beginning of the wet season, respectively. The seasonal context is already clarified in the revised manuscript (Lines 120-121): "Therefore, the measurements carried out in December and January represent the end of the dry season and the beginning of the wet season, respectively."

**Comments and suggestions:**

**Line 368, it is addressed that on clean days the photochemistry is favorable for the production of nucleated particles. however, it is also stated that there is no typical NPF cases occurred. So that means the burst of nucleation existed, but without clear growth processes. Please give more explanation about the reason.**

**Responses and Revisions:**

Good suggestion. As noted, we did observe signs of nucleation on clean days, indicating that the initial formation of nanoclusters does occur. However, we emphasize that the number concentration of sub-3

nm particles remains relatively low compared to typical NPF events. Moreover, the absence of a clear and sustained growth process suggests that these nucleated particles do not efficiently grow to larger sizes.

We propose two possible explanations for this phenomenon. First, there may be a lack of sufficient low-volatility vapors or precursors necessary to promote particle growth beyond the initial nucleation stage. Alternatively, or additionally, the newly formed particles may be rapidly lost due to coagulation with larger pre-existing particles or through condensation onto surfaces, preventing their survival and growth.

We have added further clarification to the manuscript to reflect these interpretations (Line 402-407).

"However, the number concentration of sub-3 nm particles remained relatively low compared to typical NPF events (Hong et al., 2023; Deng et al., 2022; Bianchi et al., 2016; Kulmala et al., 2013). The absence of clear growth suggests either a lack of condensable vapors to sustain particle growth or efficient loss processes such as coagulation with larger particles (Stolzenburg et al., 2023)."

**Comments and suggestions:**

***Line 405, the first sentence is not rigorous, as the authors also mentioned other studies (such as Wimmer et al., 2018 and Zhou et al., 2002) also conducted the particle size measurement below 10 nm. Please highlight the difference between this work and the others.***

**Responses and Revisions:**

Thanks for the suggestion. We have revised the related text to better highlight the unique aspects of our work. Compared to previous studies such as Wimmer et al. (2018) and Zhou et al. (2002), which conducted measurements near the ground surface, our study presents the first measurements conducted above the forest canopy in the central Amazon, providing data more representative of boundary layer conditions. Additionally, our study is the first to report sub-3 nm particle concentrations and their diurnal variation in this region.

We have revised the sentence accordingly in the manuscript (Line 444-446).

"This study is the first to provide a detailed description of the size distribution and diurnal variation of particles and ions smaller than 3 nm measured above the forest canopy in the central Amazon region, supposed to offer a more accurate representation of boundary layer conditions of the region."

***Table 1, the unit of particle number concentration should be given***

Thanks for pointing this out. We have added the unit of particle number concentration in Table 1 as suggested.

[revised manuscript text omitted]

---

## Author Comment (AC2)

**Response to Reviewer #2**

We sincerely thank the reviewer for the valuable feedback and comments, which have helped to improve our manuscript. In the following, we address the general and specific comments point by point. The reviewer's comments are given in **black bold italic**, while our responses are in normal font (non-bold, non-italic). Revised sections in the manuscript text in response to the comments are written in blue. The page and line numbers refer to the revised version. The figures and tables used in the responses are labeled the same as the revised version.

***Comments and suggestions:***

***This measurement report presents size distribution measurements of aerosol particles and ions with a particular focus on sub 10nm sizes captured at a measurement tower in the central Amazon rainforest. The measurements are taken with different state-of-the-art instruments during a period representing the wet and dry season of the region. The data is analyzed with respect to seasonal differences, diurnal variations and the influence of pollution. The paper provides valuable insights in aerosol particle concentrations and size distributions, particularly for diameters below 10 nm, for the Amazon rainforest. However, major improvements are required. I recommend to accept the paper after the following improvements have been addressed:***

**Responses and Revisions:**

Thanks for the constructive and encouraging feedback. Following the comments and suggestions of the reviewer, we have addressed all the raised points to improve the quality of our manuscript. Please find below our detailed responses to each of the comments.

***Comments and suggestions:***

***Section 2.2.4: Was the intercomparison also repeated during and/or after the measurement campaign to ensure that the instruments didn't change during the measurement campaign? Also, the presented intercomparison is only considering two of the three instruments. It would be good to have an intercomparison between all the three instruments.***

**Responses and Revisions:**

We thank the reviewer for pointing out the importance of instrument intercomparison throughout the campaign.

Regarding intercomparisons during/after the campaign between TSI and Airmodus, we were unable to perform a repeated intercomparison during or after the campaign, primarily due to malfunctions with the TSI Nano-Enhancer during our campaign. As a result, the TSI system was operated without the Nano-Enhancer throughout the measurement period, leading to a cut-off size of approximately 10 nm. This configuration prevented any overlap between the TSI system and the Airmodus PSM in the sub-10 nm range. Consequently, a full intercomparison involving both the PSM and SMPS was not feasible after the campaign.

However, to ensure the reliability of the PSM data, we followed the quality assurance procedures recommended by Lehtipalo et al. (2022). This included routine checks of the background signal, instrument stability, and proper operation protocols to maintain data integrity during the campaign.

About the intercomparison among all three instruments, we agree that a comprehensive intercomparison including the NAIS would be valuable. However, the lowest detection limit for NAIS in the particle mode is between 2 and 3 nm depending on the corona voltage as well as the properties and composition of carrier gas (Manninen et al., 2016). The intercomparison between NAIS and PSM may be strongly influenced by their lowest detection limit. Moreover, conducting a side-by-side intercomparison between TSI (Enhancer + CPC) and the NAIS also presents significant technical challenges due to the high sampling flow rate (54 L/min) of NAIS, which makes it difficult to generate high concentrations of sub-3 nm particles for parallel measurements.

We operated the NAIS by following well-established protocols and recommendations provided in Manninen et al. (2016), which includes regular monitoring of the flow rate, ion mobility spectra, and background levels, to ensure consistent performance of NAIS throughout the campaign.

***Comments and suggestions:***

***Line 209-210: The combined size distribution in Figure 2c disagree significantly for the sizes around the stitching point of NAIS and SMPS at 40 nm for most of the measurement period. For some periods the disagreement is more than one order of magnitude. The disagreement can also be seen in Table 1, where SMPS reports a median of 27 #/cm³ in the size-range 10-40 nm for "all period". However, for NAIS (particle) the size range 12-40 (i.e. the subtraction of the concentrations in the size range 2-4 and 4-12 from the concentration for the size range 2-40) reveals a value of 310, which is one order of magnitude larger than the SMPS value. Why is there such a large disagreement and which instrument is correct?***

**Responses and Revisions:**

We thank the reviewer's comments regarding the disagreement between the NAIS and SMPS measurements around the stitching point (40 nm) in Figure 2c and Table 1. We acknowledge that the particle concentrations measured by the NAIS and SMPS show notable discrepancies in the overlapping size range (between 10 and 40 nm), with the NAIS reporting notably higher values.

The large discrepancy between SMPS and NAIS measurements can be attributed to differences in their detection principles, sampling efficiencies, and particle losses. The SMPS measures particle number concentration using differential mobility analyzer (DMA) and a condensation particle counter (CPC). Specifically, the CPC used in our SMPS has a 50 % detection efficiency at approximately 10 nm, which limits its sensitivity to particles around 10 nm. Additionally, the 1 L/min inlet flow rate used during our measurements likely led to substantial diffusional losses. Although the inlet losses were calibrated using size-dependent penetration efficiency, residual uncertainties may still remain. These include potential inaccuracies in particle loss models, uncertainties in flow rate control, and environmental fluctuations that may affect diffusion rates and particle behavior (Kangasluoma et al., 2020; Von Der Weiden et al., 2009).

In contrast, the NAIS determines particle number concentration using electrometers and uses a corona charger to charge neutral particles (Mirme et al., 2007). Electrometer is highly sensitive. While this method allows detection of particles as small as 2 nm and benefits from high total flow rates (54 L/min) that reduce diffusional losses, the corona charging process is sensitive to environmental conditions such as relative humidity and aerosol composition, introducing uncertainty in the detection efficiency and size cut-off (Manninen et al., 2016).

Overall, SMPS may underestimate small particle concentrations due to diffusional losses and CPC cut-off limitations, NAIS may overestimate them due to high-sensitivity electrometer detectors and uncertainties in the corona charging process.

To further clarify, we have summarized a table comparing particle number concentrations measured by both instruments in the Amazon region, which demonstrates that our individual SMPS and NAIS results generally fall within the range of values reported in previous studies. These differences across studies may be attributed to variations in measurement periods, locations, and inlet configurations.

Table 1. Summary of particle number concentration measured with SMPS and NAIS in Amazon region

| Instrument | Size range | Measured period | Number Concentration* (cm$^{-3}$) | Literature |
|---|---|---|---|---|
| SMPS | 10-50 nm | Feb 2014 to Sept 2020 | 49 (29-81) | (Franco et al., 2022) |
| | 10-30 nm | Feb 2008 to Jul 2010 | 40 (10-121) ** | (Rizzo et al., 2018) |
| | | Nov 2012 to Oct 2014 | 89 (24-338) ** | |
| | Ultrafine | March and April 1998 | 48 (median) | (Zhou et al., 2002) |
| | 10-40 nm | Dec 2022 to Jan 2023 | 27 (13-63) | This study |
| NAIS | 2-12 nm | Jan to Jun 2014 | 473 (170-963) | (Wimmer et al., 2018) |
| | | Jul to Dec 2014 | 545 (207-1132) | |
| | 2-12 nm | Dec 2022 to Jan 2023 | 1152 (799-1570) | This study |

*The number present median values and the 25th-75th percentiles are in brackets.
** The 10th and 90th percentiles are in brackets.

It is also important to note that discrepancies around the stitching point between NAIS and SMPS are commonly shown in the literature. For instance, Figure S3(a) in Yao et al. (2018) and Figure 1 in Olin et al. (2022) both show similar discontinuities. Such inconsistencies are often related to fundamental differences in instrument design, detection principles, and uncertainties near the lower and upper detection limits of each instrument.

We have clarified this point in the revised manuscript and referred to relevant literature to help explain the observed mismatch, as shown in line 239-246.

"Additionally, in **Fig. 2c**, a noticeable difference was observed between NAIS and SMPS measurements around the stitching point (40 nm), which has also been reported in previous studies (Yao et al., 2018; Olin et al., 2022). This likely reflects differences in detection principles, instrument uncertainties, and sampling losses near their respective detection limits. SMPS may underestimate small particle concentrations due to diffusional losses and CPC cut-off limitations (Kangasluoma et al., 2020; Von Der Weiden et al., 2009), while NAIS may overestimate them due to high-sensitivity electrometer detectors and uncertainties in the corona charging process (Mirme et al., 2007; Manninen et al., 2016)."

***Comments and suggestions:***

***Line 210-211: It would be good to explain the term "Amazonian bananas" and to mention, where the "Amazonian bananas" can be seen in Figure 2c.***

**Responses and Revisions:**

Thanks for pointing this out. The term "Amazonian bananas" does not have a formal definition. In our manuscript, we adopt the terminology used by Pöhlker et al. (2018) and Franco et al. (2022), where "Amazonian bananas" refer to particle growth events that typically initiate at larger sizes, around 20 to 40 nm, rather than starting from a few nanometers as seen in classical new particle formation (NPF) events. In Figure 2c, we highlight two representative examples on December 7 (12.07) and January 20 (01.20), where such growth patterns are clearly visible. These examples are marked with dashed rectangles in the figure to emphasize the characteristic "Amazonian banana" behavior.

We have updated the manuscript accordingly (see line 218-220).

"We observed several instances of 'Amazonian bananas', characterized by particle growth initiating at a diameter between 20 and 40 nm, consistent with the observations reported by Pöhlker et al. (2018) and Franco et al. (2022). Notable examples include December 7 and January 20, which are highlighted in **Fig. 2c**."

[Figure]

Figure 2. The time series plot of (a) Wind speed and wind direction; (b) Ambient relative humidity (RH), temperature, and rainfall; (c) Particle number size distribution from the NAIS (2-40 nm, negative particle mode) and the SMPS system (40-414 nm); (d) Total number concentration of particles with diameter ranges of 1.5-1000 nm and 1.5-3.5 nm from the PSM measurement. Note that the grey parts in panels (a) and (b) and the white part in panel (c) indicate that no data are available. The two dashed rectangles in panel (c) highlight representative 'Amazonian banana' events observed on December 7 and January 20. The timestamps used here are synchronized with the local time in Manaus, called Local Time (LT), which is equal to Coordinated Universal Time (UTC) minus 4 hours.

***Comments and suggestions:***

***Line 226-227: What are the criteria to reject the data. In Figure S3, outlier with higher concentrations are also visible for relative humidity values as low as 60% - were these datapoints also rejected. Could a to restrictive rejection of such periods also by mistake reject a NPF event? It would be good to also mention how much data was rejected?***

**Responses and Revisions:**

Thanks for the valuable comments. We rejected data primarily by manually identifying periods with RH = 100% and visually inspecting the particle number size distributions for needle-like bursts, which are typically associated with rainfall events. This procedure is similar to that described in Wimmer et al. (2018), where data screening was performed by visually checking the surface plots of particle and ion size distributions. Beyond this, no additional manual rejection was applied. Data quality was assessed by following the method recommended by Manninen et al. (2016). In total, approximately 22 % of the data was excluded, when calculating the particle and ions number concentrations.

However, it's important to note that for showing the particle number size distribution in Figure 2c, we still present the full dataset, without executing the above data exclusion rule. There is little chance that we could reject a NPF event by mistake. Furthermore, if a typical NPF event had occurred, it should likely have been captured by the SMPS as well, as SMPS measurements were unaffected by precipitation events.

We have added the following sentences in the revised manuscript (see line 236-239):

"Accordingly, all data potentially affected by precipitation were carefully checked and excluded from the NAIS dataset prior to further analysis. This was done by manually identifying periods with RH = 100% and visually inspecting the particle number size distributions for needle-like burst anomalies typically associated with rainfall events."

*Comments and suggestions:*

*Line 259: December concentration for size-range "large (4-12 nm)" of 491 cm-3 does not agree with Table 1, where 497 cm-3 is written.*

**Responses and Revisions:**

Thanks for pointing out this mistake. The reviewer is correct, the value should be 497 cm$^{-3}$, as shown in Table 1. We have corrected the text accordingly in the revised manuscript (see line 288-289).

"The corresponding concentrations in December were 931 and 497 cm$^{-3}$, much higher than 558 and 306 cm$^{-3}$ observed in January."

*Comments and suggestions:*

*Line 269-271: From Dec to Jan, the median concentration of particles with diameters between 1.5 and 3.5 nm measured with PSM is increasing from 371 to 573. This contradicts the findings with the NAIS instrument, where the concentration in the intermediate (2-4nm) size range is decreasing from 931 to 558. Why is there a contradicting trend from the two instruments covering a comparable size range?*

**Responses and Revisions:**

Thanks for pointing this out. We also noticed this discrepancy between the two instruments. While we cannot provide a definitive explanation, several factors may contribute to the observed differences:

1. **Influence of the Corona Charger in NAIS Particle Mode:** The NAIS particle mode relies on a corona charger, which can affect the detection of the smallest particles. As noted by Manninen et al. (2016), the lower detection limit of NAIS is typically between 2 and 3 nm, depending on the corona voltage and the composition of the carrier gas (i.e., ambient conditions). In our study,

the reported 2-4 nm size range may be influenced by this limitation, particularly under high-humidity conditions.

2. **Differences in Sizing Principles:** NAIS measures the electrical mobility diameter, whereas the PSM determines size based on condensation activation-specifically, the Kelvin equivalent diameter. This fundamental difference in measurement principles can lead to discrepancies in reported concentrations for similar nominal size ranges.

Additionally, it is worth noting that the ion mode of NAIS, which is not affected by the corona charger, shows slightly higher concentrations in January, consistent with the data trend revealed by PSM.

We have further explained these two points in the revised manuscript in line 170-173:

"The particle mode of NAIS operates using a corona charger, which can potentially influence the detection of the smallest particles. Moreover, the lower detection limit typically ranging between 2 and 3 nm also depends on corona voltage and ambient gas composition (Manninen et al., 2016). Usually, the results for particles below 2 nm were not used for data analysis."

And in line 226-228:

"It should be noted that, due to their differing measurement principles, the NAIS reports electrical mobility diameters, whereas the PSM measures condensation activation diameters (Kelvin equivalent sizes)."

***Comments and suggestions: Line 284-285: Where is the number concentration of 397 cm-3 coming from? It is not one of the values from the Table 1. It should be better explained, how this value was derived. Also: Table S1 lists two values 450 and 610 labeled as "this study". How are these values derived, since they also differ from the value mentioned in Table 1?***

**Responses and Revisions:**

Thanks for pointing this out. We apologize for the inconsistency. The value of 397 cm$^{-3}$ stated on line 285 was incorrect. The correct number concentration is 491 cm$^{-3}$, as reported in Table 1. Additionally, the concentrations for December and January listed in Table S1 have now been updated to 371 and 573 cm$^{-3}$, respectively. We have corrected these values in the manuscript and carefully double-checked all related values throughout the text and supplementary materials to ensure consistency and accuracy.

The corresponding correction in the revised manuscript in line 313-315 is:

"Under pristine conditions in the Amazon region, the particle number concentration (491 cm$^{-3}$) was notably lower than in megacities (> 8500 cm$^{-3}$) such as Nanjing, Shanghai and San Pietro Capo Fiume (Xiao et al., 2015; Kontkanen et al., 2017)."

**Table S1.** The summary of the sub-3 nm particle number concentration measurements using PSM across different global environments

| Location | Time period | Size range (nm) | Concentration* ($cm^{-3}$) | Environmental type | Literature |
|---|---|---|---|---|---|
| Helsinki, Finland | Jan 2015 – Dec 2015 | 1.1 – 3.0 | 6.0E+03 | Urban, city | (Kontkanen et al., 2017) |
| Nanjing, China | Dec 2014 –Jan 2015 | 1.1 – 3.0 | 1.7E+04 | Urban, city | (Qi et al., 2015) |
| Shanghai, China | Nov 2013 –Jan 2014 | 1.3 – 3.0 | 8.5E+03 | Urban, city | (Xiao et al., 2015) |
| San Pietro Capo Fiume, Italy | Jun 2012 –Jul 2012 | 1.5 – 3.0 | 8.5E+03 | Urban, city | (Kontkanen et al., 2016) |
| Centreville, US | Jun 2013 – Jul 2013 | 1.1 – 2.1 | 5.9E+02 | Urban, farm/forest | (Yu et al., 2014) |
| Kent, US | Dec 2011 – Jan 2012 | 1.3 – 3.0 | 4.7E+02 | Urban, town | |
| Brookhaven, US | Jul 2011 – Aug 2011 | 1.3 – 3.0 | 8.0E+02 | Rural, coast | |
| Puy de Dôme, France | Jan 2012 – Feb 2012 | 1.3 – 2.5 | 5.0E+02 | Rural, mountain | (Rose et al., 2015) |
| Hyytiälä, Finland | Aug 2010; Mar 2011 – Apr 2011; Aug 2011 – Sep 2011; Apr 2012 – May 2012; May 2013 – Jul 2013; May 2015 – Apr 2016 | (1.1/1.3) – 3.0 | 1.6E+03 | Rural, forest | (Kontkanen et al., 2017) |
| | April 2014 – April 2020 | 1.1 – 2.5 | 4.1E+02 | Rural, forest | (Sulo et al., 2021) |
| Amazonian, Brazil | Dec 2022 | 1.5 – 3.5 | 3.7E+02 | Rural, forest | This study |
| Amazonian, Brazil | Jan 2023 | 1.5 – 3.5 | 5.7E+02 | Rural, forest | This study |

* Median values

*Comments and suggestions:*

*Line 329-330: The described trend of the particles larger than 15 nm (D15-40nm) is very hard to see in the plot - especially for the Dec. period. The scale for the colormap starts at 1e1. However, the colors between 1e1 and 1e2 are not present in the plot. Starting the color scale at 1e2 would help to make it easier to see the mentioned trends.*

**Responses and Revisions:**

Good suggestion. We agree that the trends of particles in the 15-40 nm range during the December period were not clearly visible due to the colormap scale. Following the reviewer's suggestion, we have adjusted the lower bound of the color scale from 1e1 to 1e2 in the revised Fig. 7 (Lines 497-500).

[Figure]

Figure 7. Diurnal variation of the particle number size distribution measured by NAIS during December 2022 (left panel) and January 2023 (right panel). The plot data represent the median values.

Instead of presenting the nucleation mode as a whole, we divide the particles into three size categories: 2-6 nm, 6-20 nm, and 20-40 nm. Similar multimodal particle size distributions have also been reported in previous studies (Zhou et al., 2002; Rissler et al., 2004). To better illustrate the diurnal variation across these size ranges, in the revised Supporting Information we additionally show the diurnal variation of the total particle number concentration within each category, as presented below.

[Figure]

**Figure S5.** Diurnal variation of particle number concentrations in the 2-6 nm (blue), 6-20 nm (orange), and 20-40 nm (green) size ranges measured by NAIS during December 2022 (left panel) and January 2023 (right panel). Note: particle number concentrations for the 6-20 nm and 20-40 nm ranges have been multiplied by a factor of 3 for better visualization.

The corresponding correction in the revised manuscript in line 358-368 is as follows:

"To better illustrate the dynamic changes in particle number concentrations, we separately analyzed particles in the 2-6 nm, 6-20 nm, and 20-40 nm size ranges. Multimodal particle size distributions that correspond to these sizes have been reported in previous studies (Rissler et al., 2004; Zhou et al., 2002). The number concentration for 2-6 nm particles steadily increased throughout the daytime, peaking in the late afternoon for both December and January (see also in Figure S5). Rissler et al. (2004) also observed a pronounced nucleation mode around 16:00-18:00 local time, followed by a decrease and a secondary rise around 06:00-07:00 in the early morning. In December, the concentration of 6-20 nm particles also increased notably from 08:00 in the morning to the afternoon, shortly after the initial rise of 2-6 nm particles. The phenomenon of lagged increasing suggests particle growth processes. In January, particles in the 20-40 nm size range exhibited a continuous increase during nighttime (18:00-24:00), reaching a maximum in the early morning hours. This behavior, however, was not observed in December."

*Comments and suggestions:*

*Line 353-354: Where does the BC mass concentration come from and how was it measured? More explanation is required.*

**Responses and Revisions:**

Thanks for pointing this out. Clarification sentences have been added to the revised manuscript. The equivalent black carbon (BC) mass concentration was obtained from a Multi-angle Absorption Photometer (MAAP, model 5012, Thermo Scientific), which has been operated as part of the long-term aerosol measurements at the ATTO site (Pöhlker et al., 2018; Saturno et al., 2018). We have now clarified this point in the revised manuscript.

Sentences to add into the manuscript (Line 129-132):

"The equivalent black carbon mass concentrations were obtained from a Multi-Angle Absorption Photometer (MAAP, model 5012, Thermo Electron Group), which is part of the long-term aerosol monitoring program at the ATTO site (Pöhlker et al., 2018; Saturno et al., 2018)."

*Comments and suggestions:*

*Line 374–379: "plume" should be changed to "polluted" and "clear" should be changed to "clean"*

**Responses and Revisions:**

Thanks for the suggestion. We have revised the wording by replacing "plume" with "polluted" and "clear" with "clean" to improve clarity and consistency in terminology (Line 411-412 and Line 414-415).

"**Figure 10** illustrates the diurnal patterns of the particle size distributions on clean and polluted days. As shown in the left panel of **Fig. 10**, the number of nucleation mode particles is higher on clean days."

"The diurnal variation observed during polluted days (the right panel of **Fig. 10**) exhibited a pattern similar to that described in **Fig. 7**."

*Comments and suggestions:*

*Line 374-375: It would be good to indicate the time of the large number of nucleation mode particles, since there is also a period with a very low nucleation mode number concentration from approx. 05:00 to 11:00. For Figure 10, also the color scale starts at 1e1. However, a min. value of 1e2 would most likely help to see features and trends better.*

**Responses and Revisions:**

Good suggestion. Instead of showing only the nucleation mode as a whole, we present two particle size categories: 2-6 nm and 6-20 nm. For particles in the 2-6 nm size range, we observed a noticeable concentration increase starting around 08:00 during clean days. For particles in the 6-20 nm size range, both size and concentration increase midday, with a more pronounced enhancement during clean days.

To better illustrate the diurnal variation within these particle size ranges, in the revised Supporting Information we additionally show the diurnal variation of the total particle number concentration for the 2-6 nm, 6-20 nm, and 20-40 nm size ranges, as shown below.

[Figure]

**Figure S6**. Diurnal variation of particle number concentrations in the 2-6 nm (blue), 6-20 nm (orange), and 20-40 nm (green) size ranges measured by NAIS during clean days (left panel) and polluted days (right panel). Note: particle number concentrations for the 6-20 nm and 20-40 nm ranges have been multiplied by a factor of 3 for better visualization.

The sentence has been added in the revised manuscript (Line 411-414):

"Figure 10 illustrates the diurnal patterns of the particle size distributions on clean and polluted days. As shown in the left panel of Fig. 10, the number of nucleation mode particles is higher on clean days. Specifically, the particle concentrations in the 2-6 nm range started to increase around 08:00 (see also in **Figure S6**), and 6-20 nm particles showed growth and elevated concentrations in the afternoon."

As suggested, we have also adjusted the minimum value of the color scale in Figure 10 from 1e1 to 1e2 to enhance the visibility of temporal features and trends (Line 512-515).

[Figure]

Figure 10. Diurnal variation of the particle number size distribution measured by NAIS during clean days (left panel) and polluted days (right panel). The plot data represents the median values.

***Comments and suggestions:***

***Line 375-376: "Figure 10a" should be changed to "left panel of Figure 10" and "Fig. 10b" should be changed to "right panel of Figure 10".***

**Responses and Revisions:**

Thanks for pointing this out. We have revised the manuscript accordingly.

The related modifications in the revised manuscript are in Line 411-412 and Line 414-415:

"As shown in the left panel of **Fig. 10**, the number of nucleation mode particles is higher on clean days."

"The diurnal variation observed during polluted days (the right panel of **Fig. 10**) exhibited a pattern similar to that described in Fig. 7."

***Comments and suggestions:***

***Line 405-407: In the "Results and discussion" chapter, many of the sub 10nm findings of this study are compared to measurements reported in the literature. It would be good to specify more precisely which aspect of the paper is the "first-time" presentation.***

**Responses and Revisions:**

Thanks for pointing this out.

Several previous studies have reported particle number size distributions below 10 nm in the Amazon region (Wimmer et al., 2018; Rissler et al., 2004; Zhou et al., 2002). However, our measurements are the first example conducted within the forest and above the canopy, providing a more representative characterization of boundary layer conditions. Furthermore, to our knowledge, this is the first study to report sub-3 nm particle concentrations under such pristine conditions in the central Amazon region.

In the revised manuscript, we have added statements to clarify the novelty of this study and emphasize the uniqueness of the measurement results (Line 444-446):

"This study is the first to provide a detailed description of the size distribution and diurnal variation of particles and ions smaller than 3 nm measured above the forest canopy in the central Amazon region, supposed to offer a more accurate representation of boundary layer conditions of the region."

***Comments and suggestions:***

***Table 1: Since the sizes represented by the different instruments are not all comparable (as mentioned in 217-218) it would be good to indicate, that the sizes reported for PSM are different compared to the sizes reported by the other listed instruments.***

**Responses and Revisions:**

Thanks for the valuable suggestion. We agree that the particle size ranges reported by the different instruments are not directly comparable due to their distinct measurement principles. We have therefore clarified this in the caption of Table 1 by explicitly indicating that the sizes reported for the PSM refer to condensation activation diameters (Kelvin equivalent), while the sizes for the other instruments represent electrical mobility diameters. This distinction has also been briefly explained in Section 3.1 to avoid any confusion for the readers.

One sentence has been added in the revised manuscript below Table 1 (Line 458-459):

"* PSM measures the condensation activation diameters (Kelvin equivalent), while the sizes reported by the other instruments represent electrical mobility diameters."

Another sentence has also been included in the main text of the revised manuscript (Line 226-228):

[revised manuscript text omitted]